# SEGNO: Generalizing Equivariant Graph Neural Networks with Physical Inductive Biases

**Yang Liu**[1,2][*], **Jiashun Cheng**[1,2][*], **Haihong Zhao**[1], **Tingyang Xu**[3], **Peilin Zhao**[3], **Fugee Tsung**[1,2],
**Jia Li**[1,2][†], **Yu Rong**[3][†]
[1]The Hong Kong University of Science and Technology (Guangzhou)
[2]The Hong Kong University of Science and Technology   [3]Tencent AI Lab

## Abstract

Graph Neural Networks (GNNs) with equivariant properties have emerged as powerful tools for modeling complex dynamics of multi-object physical systems. However, their generalization ability is limited by the inadequate consideration of physical inductive biases: (1) Existing studies overlook the continuity of transitions among system states, opting to employ several discrete transformation layers to learn the direct mapping between two adjacent states; (2) Most models only account for first-order velocity information, despite the fact that many physical systems are governed by second-order motion laws. To incorporate these inductive biases, we propose the **S**econd-order **E**quivariant **G**raph **N**eural **O**rdinary Differential Equation (SEGNO). Specifically, we show how the second-order continuity can be incorporated into GNNs while maintaining the equivariant property. Furthermore, we offer theoretical insights into SEGNO, highlighting that it can learn a unique trajectory between adjacent states, which is crucial for model generalization. Additionally, we prove that the discrepancy between this learned trajectory of SEGNO and the true trajectory is bounded. Extensive experiments on complex dynamical systems including molecular dynamics and motion capture demonstrate that our model yields a significant improvement over the state-of-the-art baselines.

## 1 Introduction

Equivariant Graph Neural Networks (Equiv-GNNs) (Satorras et al., 2021; Han et al., 2022b; Brandstetter et al., 2021; Huang et al., 2022; Wu et al., 2024) have emerged as essential tools for simulating the multi-object physical system, i.e., N-body systems, which is relevant to numerous fundamental scientific domains, including molecular dynamics (Karplus & McCammon, 2002), protein folding (Gligorijević et al., 2021), robot motion planning/control (Siciliano et al., 2009). Specifically, given the input state, they learn to predict the output state after a specific timestep. To achieve these, Equiv-GNNs model the whole system as a geometric graph, which treats physical objects as nodes, and physical relations as edges, and encode the symmetry into a message-passing network to ensure their outputs are equivariant with respect to any translation/orientation/reflection of the inputs. Such property makes them well-suited

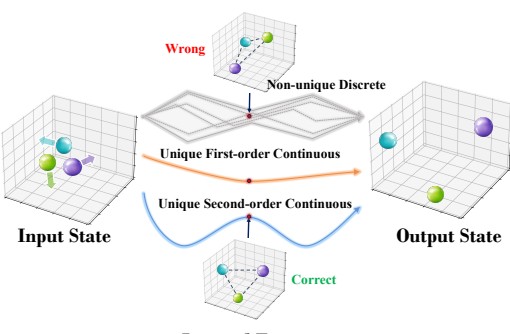

Figure 1: Learned trajectories of models with different inductive bias. All models can map input to output. However, discrete and first-order continuous models fail to learn the true intermediate states due to the lack of considering continuity and second-order laws.

for learning the unknown dynamics of complex physical systems that cannot be described analytically (Han et al., 2022a;b).

---

[*]Equal contribution. Work is done when Yang Liu and Jiashun Cheng worked as interns in Tencent AI Lab.
[†]Corresponding authors (jialee@ust.hk, yu.rong@hotmail.com)

Learning to model such interacting systems is challenging. Given the vast parameter space of GNNs and finite observations of transitions of system states, there would exist multiple solutions that satisfy observed data (Curry & Morgan, 2006). Therefore, learning the real dynamic function from these solutions is crucial to model generalization ability Gruver et al. (2022). Though Equiv-GNNs (Han et al., 2024) have partially addressed this challenge by eliminating models that lack symmetry, existing Equiv-GNNs have yet to incorporate sufficient physical inductive bias to model the physical dynamics.

In particular, two essential inductive biases have not been well investigated in this field. First, existing models (Satorras et al., 2021; Brandstetter et al., 2021) are composed of a sequence of discrete state transformation layers, which learn a direct-mapping between adjacent states with discrete trajectories. We refer to them as *discrete models*. They are inconsistent with the continuous nature of system trajectories and fail to learn correct intermediate states. Second, most models only account for first-order information. Many physical dynamical systems, such as Newton's equations of motion, are governed by second-order laws (Norcliffe et al., 2020). Therefore, these methods learn incomplete representations of the system's state and fail to capture the underlying dynamics of the physical systems. In Figure 1, we illustrate the comparison of learned trajectories of models with different types of inductive bias.

In this work, we take a deep insight into the continuity and second-order inductive bias in Equiv-GNNs and propose a framework dubbed **S**econd-order **E**quivariant **G**raph **N**eural **O**rdinary Differential Equation (SEGNO[1]). Different from previous models that use Equiv-GNNs to fit discrete kinematic states, SEGNO introduces Neural Ordinary Differential Equations (Neural ODE) to approximate a continuous trajectory between two observed states. Furthermore, to better estimate the underlying dynamics, SEGNO is built upon second-order motion equations to update the position and velocity of the physical systems. Theoretically, we prove the uniqueness of the learned latent trajectory of SEGNO and further provide an upper bound on the discrepancy between the learned and the actual latent trajectory. Meanwhile, we prove that SEGNO can maintain equivariance properties identical to the backbone Equiv-GNNs. This property offers the flexibility to adapt various backbones in SEGNO to suit different downstream tasks in plug-and-play manner. We conduct extensive experiments on both synthetic and real-world physical systems. Our results reveal that SEGNO has a better generalization ability over the state-of-the-art baselines and second-order inductive bias is beneficial to learn complex multi-object dynamics.

## 2 BACKGROUND

**N-body System**   We study N-body systems (Kipf et al., 2018; Huang et al., 2022) with a set of $N$ particles $\mathcal{P} = \{P_i\}_{i=1}^{N}$. At time $t$, the state of each particle in the system is represented by:
**1.** geometric features including the position vector $\boldsymbol{q}_i^{(t)} \in \mathbb{R}^3$ and the velocity vector $\dot{\boldsymbol{q}}_i^{(t)} \in \mathbb{R}^3$;
**2.** non-geometric features such as mass or charge, denoted by $\boldsymbol{h}_i \in \mathbb{R}^d$; **3.** spatial connection with others, where an edge $e_{ij}$ is constructed via geometric distance cutoff or physical interaction (e.g., chemical bonds) and the edge attributes (e.g., object distances, bond type) are denoted by $a_{ij}$. For simplicity, we denote $(\boldsymbol{q}^{(t)}, \dot{\boldsymbol{q}}^{(t)})$ and $(\boldsymbol{h}, \boldsymbol{e} = \{e_{ij}\}, \boldsymbol{a} = \{a_{ij}\})$ as dynamic and static state information at system level correspondingly. In this work, we focus on dynamical systems that can be formulated as:

$$\ddot{\boldsymbol{q}}^{(t)} = \frac{d^2\boldsymbol{q}^{(t)}}{dt^2} = f(\boldsymbol{q}^{(t)}, \boldsymbol{h}), \tag{1}$$

where $\ddot{\boldsymbol{q}}^{(t)}$ is the acceleration at time $t$. Given a trajectory $\boldsymbol{q}$[2] with the initial system states $(\boldsymbol{q}^{(t_0)}, \dot{\boldsymbol{q}}^{(t_0)})$ at time $t_0$ and static states $(\boldsymbol{h}, \boldsymbol{e}, \boldsymbol{a})$, our goal is to predict the subsequent position $\boldsymbol{q}^{(t_1)}$ within a fixed time interval $T = t_1 - t_0$.

$E(3)$ **Equivariance**   In 3-dimensional Euclidean space, the laws of physics remain invariant regardless of $E(3)$ transformations, including translation, rotation, and reflection. Formally, a function $\mathcal{F} : \boldsymbol{X} \times \boldsymbol{P} \to \boldsymbol{Y}$, where $\boldsymbol{X}, \boldsymbol{Y} \subset \mathbb{R}^3$, is $E(3)$-equivariant, if for any transformation $o \in E(3)$,

$$\mathcal{F}(o \circ \boldsymbol{x}, \cdots) = o \circ \mathcal{F}(\boldsymbol{x}, \cdots), \quad \boldsymbol{x} \in \boldsymbol{X}. \tag{2}$$

---

[1]SEGNO is also a musical term in Italian meaning "sign", marking the beginning or end of a musical repeat.

[2]To avoid ambiguity, $\boldsymbol{q}_i^{(t)}$ denotes the position of the $i$-th particle at time $t$, whereas $\boldsymbol{q}$ denotes the trajectory over the entire time interval.

**Equivarant GNNs** In general, given the system state at time $t_0$, a modern Equiv-GNN $\eta_\theta$ with parameters $\theta$ directly predicts $\boldsymbol{q}^{(t_1)}$ by leveraging several message passing layers on an interaction graph. To maintain brevity and avoid ambiguity, we omit the temporal superscript $t$ here since the predictions at different times share the same model $\eta_\theta$. Specifically, each layer of $\eta_\theta$ computes

$$\boldsymbol{m}_{ij} = \mu(\boldsymbol{q}_i, \boldsymbol{q}_j, \dot{\boldsymbol{q}}_i, \dot{\boldsymbol{q}}_j, \boldsymbol{h}_i, \boldsymbol{h}_j, a_{ij}), \quad \boldsymbol{q}'_i, \dot{\boldsymbol{q}}'_i, \boldsymbol{h}'_i = \nu(\boldsymbol{q}_i, \dot{\boldsymbol{q}}_i, \boldsymbol{h}_i, \sum_{j \in \mathcal{N}_i} \boldsymbol{m_{ij}}), \tag{3}$$

where $\mu$ and $\nu$ are the edge message function and node update function, respectively, $\boldsymbol{m}_{ij}$ defines the message between node $i$ and $j$. $\mathcal{N}_i$ collects the neighbors of node $i$. The prediction is obtained by applying several iterations of message passing. To construct equivariant layers, $\mu$ and $\nu$ could be both equivariant (e.g., SEGNN (Brandstetter et al., 2021)) or alternatively, $\mu$ is equivariant and $\nu$ is invariant (e.g., EGNN (Satorras et al., 2021) and GMN (Huang et al., 2022)).

## 3 SEGNO FRAMEWORK

In this section, we introduce how the proposed SEGNO works and its equivariant properties. In our dynamic systems, we incorporate the ODE formulation to model the latent continuous trajectory with initial system states $(\boldsymbol{q}^{(t_0)}, \dot{\boldsymbol{q}}^{(t_0)})$. The position $\boldsymbol{q}^{(t_0+t')}$ for any $t' \in [0, T]$ is calculated as

$$\phi_{t',g}(\boldsymbol{q}^{(t_0)}) := \boldsymbol{q}^{(t_0+t')} = \boldsymbol{q}^{(t_0)} + \int_{t_0}^{t_0+t'} \left( \dot{\boldsymbol{q}}^{(t_0)} + \int_{t_0}^{t} f(\boldsymbol{q}^{(m)}, \boldsymbol{h}) \, dm \right) dt$$

$$= \boldsymbol{q}^{(t_0)} + \int_{t_0}^{t_0+t'} g(\boldsymbol{q}^{(t)}, \boldsymbol{h}) \, dt, \tag{4}$$

where $g$ represents a mapping from the trajectory to its first-order derivative $\dot{\boldsymbol{q}}^{(t)}$. In this vein, we can denote $\boldsymbol{q}^{(t_1)}$ as $\phi_{T,g}(\boldsymbol{q}^{(t_0)})$. Note that most physical dynamical systems follow the second-order motion law. The velocity $\dot{\boldsymbol{q}}^{(t_0+t')}$ for any $t' \in [0, T]$ can further be computed as

$$\psi_{t',g,f}(\boldsymbol{q}^{(t_0)}) := \dot{\boldsymbol{q}}^{(t_0+t')} = g(\boldsymbol{q}^{(t_0)}, \boldsymbol{h}) + \int_{t_0}^{t_0+t'} f(\boldsymbol{q}^{(t)}, \boldsymbol{h}) \, dt. \tag{5}$$

To incorporate the second-order inductive bias, SEGNO parameterizes the acceleration function

$$\ddot{\boldsymbol{q}}_\theta^{(t)} = f_\theta(\boldsymbol{q}^{(t)}, \boldsymbol{h}), \tag{6}$$

where $f_\theta$, an approximation of $f$, represents an Equiv-GNN with parameters $\theta$ which computes:

$$\boldsymbol{m}_{ij}^{(t)} = \mu(\boldsymbol{q}_i^{(t)}, \boldsymbol{q}_j^{(t)}, \boldsymbol{h}_i, \boldsymbol{h}_j, a_{ij}), \quad \ddot{\boldsymbol{q}}_{\theta,i}^{(t)}, \boldsymbol{h}_i = \nu(\boldsymbol{q}_i^{(t)}, \boldsymbol{h}_i, \sum_{j \in \mathcal{N}_i} \boldsymbol{m}_{ij}^{(t)}), \tag{7}$$

where $\mu$ and $\nu$ are determined by this Equiv-GNN backbone. Let $\boldsymbol{q}_\theta$ denote the approximated trajectory generated by SEGNO satisfying the initial conditions $\boldsymbol{q}_\theta^{(t_0)} = \boldsymbol{q}^{(t_0)}, \dot{\boldsymbol{q}}_\theta^{(t_0)} = \dot{\boldsymbol{q}}^{(t_0)}$. Then, following Eq. 4, the predicted position of SEGNO at time $t_1$ can be represented by

$$\phi_{T,g_\theta}(\boldsymbol{q}^{(t_0)}) = \boldsymbol{q}_\theta^{(t_1)} = \boldsymbol{q}^{(t_0)} + \int_{t_0}^{t_0+T} g_\theta(\boldsymbol{q}^{(t)}, \boldsymbol{h}) \, dt$$

$$= \boldsymbol{q}_\theta^{(t_0)} + \int_{t_0}^{t_0+T} \left( \dot{\boldsymbol{q}}_\theta^{(t_0)} + \int_{t_0}^{t} f_\theta(\boldsymbol{q}^{(m)}, \boldsymbol{h}) \, dm \right) dt \tag{8}$$

$$= \boldsymbol{q}_\theta^{(t_0)} + \int_{t_0}^{t_0+T} \psi_{t,g_\theta,f_\theta}(\boldsymbol{q}_\theta^{(t_0)}) \, dt,$$

where $g_\theta$, a parameterized version of $g$, is determined by the integral of the GNN $f_\theta$ in Eq. 4. Since it is infeasible to directly calculate the integration in $\phi_{T,g_\theta}(\boldsymbol{q}^{(t_0)})$ and $\psi_{T,g_\theta,f_\theta}(\boldsymbol{q}_\theta^{(t_0)})$ with parameterized $g_\theta, f_\theta$, in SEGNO, we utilize an ODE solver to generate a discrete trajectory that serves as an approximation of the latent continuous trajectory. Specifically, we divide the entire time interval $T$ into $\tau$ equal sub-intervals with timestep $\Delta t = T/\tau$. We then denote $\Psi_{\Delta t,g,f}$ and $\Phi_{\Delta t,g}$ as the numerical integrators that approach $\psi_{\Delta t,g,f}$ and $\phi_{\Delta t,g}$ using the following equations

$$\dot{\boldsymbol{q}}_\theta^{(t+\Delta t)} = g_\theta(\boldsymbol{q}_\theta^{(t+\Delta t)}, \boldsymbol{h}) = \Psi_{\Delta t,g_\theta,f_\theta}(\boldsymbol{q}_\theta^{(t)}) = \dot{\boldsymbol{q}}_\theta^{(t)} + \mathcal{G}_1(f_\theta(\boldsymbol{q}_\theta^{(t)}, \boldsymbol{h}), \Delta t),$$

$$\boldsymbol{q}_\theta^{(t+\Delta t)} = \Phi_{\Delta t,g_\theta}(\boldsymbol{q}_\theta^{(t)}) = \boldsymbol{q}_\theta^{(t)} + \mathcal{G}_2(\Psi_{\Delta t,g_\theta,f_\theta}(\boldsymbol{q}_\theta^{(t)}), \Delta t), \tag{9}$$

where $\mathcal{G}_1$ and $\mathcal{G}_2$ are the increment functions that approximate the increment of a continuous integral given the initial value of the integrand and the integration width $\Delta t$. For instance, with the increment functions $\mathcal{G}_1(x, y) = \mathcal{G}_2(x, y) = x \times y$, the numerical integrators become the Euler integrators

$$\dot{\boldsymbol{q}}_\theta^{(t+\Delta t)} = \dot{\boldsymbol{q}}_\theta^{(t)} + f_\theta(\boldsymbol{q}_\theta^{(t)})\Delta t, \quad \boldsymbol{q}_\theta^{(t+\Delta t)} = \boldsymbol{q}_\theta^{(t)} + \dot{\boldsymbol{q}}_\theta^{(t+\Delta t)}\Delta t. \tag{10}$$

For approximation, SEGNO composites $\tau$ integrator $\Phi_{\Delta t, g_\theta}$ as a Neural ODE solver following

$$\boldsymbol{q}_\theta^{(t_1)} = \phi_{T, g_\theta}(\boldsymbol{q}^{(t_0)}) := \Phi_{\Delta t, g_\theta} \circ \cdots \circ \Phi_{\Delta t, g_\theta}(\boldsymbol{q}^{(t_0)}) = (\Phi_{\Delta t, g_\theta})^\tau(\boldsymbol{q}^{(t_0)}). \tag{11}$$

As a result, SEGNO offers a way to reuse existing Equiv-GNNs to build second-order Neural ODEs.

However, a natural question arises: *does a Neural ODE solver compromise the equivariance of backbone GNNs?* To address this question, we show that the equivariance property of backbone GNNs could be maintained in SEGNO.

**Proposition 3.1.** *Suppose the backbone GNN $f_\theta$ of SEGNO is $O(3)$-equivariant and translation-invariant, and the integrators' increment function $\mathcal{G}_1, \mathcal{G}_2$ are $O(3)$-equivariant, then the output trajectory $\boldsymbol{q}_\theta$ is $E(3)$-equivariant.*

The proof and example illustrations are provided in Appendix A.1, where we show that the general numerical integrators including symplectic Euler, Velocity Verlet, and Leapfrog satisfy Proposition 3.1. Meanwhile, since Equiv-GNNs such as EGNN (Satorras et al., 2021) and GMN (Huang et al., 2022) are built upon $O(3)$-equivariant and translation-invariant functions, SEGNO would preserve the same equivariant property as the backbone GNNs.

## 4 SEGNO ANALYSIS

Besides equivariance, another essential problem is how SEGNO learns from observed system states. In this section, we examine the approximation quality of SEGNO.

### 4.1 SOLUTION UNIQUENESS

It is known that continuous dynamics have a unique solution under specific continuous conditions, according to Picard's existence theorem (Coddington et al., 1956).

**Lemma 4.1.** *For the system $\ddot{\boldsymbol{q}}^{(t)} = f(\boldsymbol{q}^{(t)}, \boldsymbol{h})$, with given initial position $\boldsymbol{q}^{(t_0)}$ and velocity $\dot{\boldsymbol{q}}^{(t_0)}$, if $f$ is Lipschitz continuous, then this system has a unique solution $\boldsymbol{q}^{(t)}$ over the interval $t \in [t_0, t_1]$.*

The proof is provided in Appendix A.2. In addition, under the SEGNO framework, we can obtain the following results:

**Proposition 4.2.** *Given the same conditions as in Lemma 4.1, if the realistic measurement on $\boldsymbol{q}^{(t_1)}$ is given, there exists a $f_{\theta^*}$ obtained by minimizing the discrepancy between $\boldsymbol{q}_{\theta^*}^{(t_1)}$ and $\boldsymbol{q}^{(t_1)}$, such that $f_{\theta^*}(\boldsymbol{q}_{\theta^*}^{(t)}, \boldsymbol{h}) = f(\boldsymbol{q}^{(t)}, \boldsymbol{h})$ holds over the interval $t \in [t_0, t_1]$.*

The detailed proof is given in Appendix A.3. This proposition remarks that SEGNO can be trained in line with prior studies (Satorras et al., 2021; Brandstetter et al., 2021). That is, given $t_0$ and $t_1$ as the input and target timesteps, SEGNO is trained to minimize the discrepancy between the exact and approximated positions:

$$\mathcal{L}_{\text{train}} = \sum_{s \in \mathcal{D}_{\text{train}}} ||\boldsymbol{q}_{\theta,s}^{(t_1)} - \boldsymbol{q}_s^{(t_1)}||_2, \tag{12}$$

where $\mathcal{D}_{\text{train}}$ denotes the training set. With a slight abuse of notation, here we denote $\boldsymbol{q}_{\theta,s}^{(t_1)}, \boldsymbol{q}_s^{(t_1)}$ as the model prediction and actual position of trajectory $s$. If our learned model adequately approximates the system, Proposition 4.2 shows that it becomes possible for SEGNO to recover the latent trajectories of $[t_0, t_1]$ between input and output system states via Neural ODE. In contrast, in the absence of continuous constraints, multiple discrete trajectories can exist between the input and output, making it challenging for discrete models to accurately learn the underlying dynamic functions.

## 4.2 APPROXIMATION ABILITY

In this section, we theoretically show that SEGNO is capable of learning a trajectory that remains bounded to the real solution. To achieve this, we first derive the boundness of the learned second-order derivative $f_\theta$. Existing theoretical findings (Zhu et al., 2022) have illustrated that training using Neural ODE solvers results in a bounded approximation of their first-order derivatives. In this work, we generalize this theorem to the second-order Neural ODE employed with Euler integrator.

### 4.2.1 BOUND OF ACCELERATION

To begin with, we first introduce relevant notations. Let $\mathcal{B}(\boldsymbol{p}, r) \subset \mathbb{R}^3$ denotes a real ball of radius $r > 0$ centered at $\boldsymbol{p} \in \mathbb{R}^3$, for a subset $\mathcal{K} \subset \mathbb{R}^3$ and an analytic function $\eta$, we define

$$\mathcal{B}(\mathcal{K}, r) = \bigcup_{\boldsymbol{p} \in \mathcal{K}} \mathcal{B}(\boldsymbol{p}, r), \quad ||\eta||_\mathcal{K} = \sup_{\boldsymbol{p} \in \mathcal{K}} ||\eta(\boldsymbol{p})||_\infty. \tag{13}$$

We denote a set of points on the exact trajectory associated with the time increment interval $[a, b]$ as

$$V_a^b = \{\phi_{t',g}(\boldsymbol{q}^{(t_0)})|a \leq t' \leq b\}. \tag{14}$$

Given the time interval $T = t_1 - t_0$ uniformly partitioned into $\tau$ sub-intervals of timestep $\Delta t$ (i.e., $t_0 < t_0 + \Delta t < \cdots < t_1$), the following theorem holds:

**Theorem 4.3.** *For $r_1, r_2 > 0$, the ODE solver is $\tau$ compositions of an Euler Integrator $\Phi_{\Delta t, g_\theta}$ with $\Delta t = T/\tau$, and we denote the supremum norm of the approximation error for the ODE solver within $\mathcal{B}(V_0^{2T}, r_1)$ as*

$$\mathcal{L}_0^{2T} := \frac{1}{T}||(\Phi_{\Delta t, g_\theta})^\tau - \phi_{T,g}||_{\mathcal{B}(V_0^{2T}, r_1)}.$$

*Suppose that the target $f$ and the learned $f_\theta$ are analytic and bounded by $m_2$ on $\mathcal{B}(V_0^{2T}, r_2)$, and the target $g$ and the learned $g_\theta$ are analytic and bounded by $m_1$ on $\mathcal{B}(V_0^{2T}, r_1)$. Then, there exist constants $T_0$ such that, if $0 < T < T_0$, $\forall t \in [t_0, t_1]$,*

$$||f_\theta(\boldsymbol{q}^{(t)}, \boldsymbol{h}) - f(\boldsymbol{q}^{(t)}, \boldsymbol{h})||_\infty \leq O(\Delta t + \frac{\mathcal{L}_0^{2T}}{\Delta t}).$$

The proof is reported in Appendix A.4. The assumption that the true $f, g$ and the learned $f_\theta, g_\theta$ are analytic and bounded within specific balls is valid and grounded in real-world physical observations. Concretely, the trajectories of most dynamic physical systems exhibit smoothness, implying that there are no sudden changes in velocity or acceleration. This smooth property ensures that the system behaves in a physically realistic and predictable manner, which is a widely employed practice in control theory and dynamic systems analysis (Zhu et al., 2022).

### 4.2.2 BOUND OF TRAJECTORY

Based on Theorem 4.3, we can now analyze the error introduced by the Euler integrator. We introduce two metrics common in classical numerical analysis, namely, local and global truncation error (Poli et al., 2020). The local truncation error $\epsilon_{t+\Delta t}$ of SEGNO is defined as:

$$\epsilon_{t+\Delta t} = ||\boldsymbol{q}^{(t+\Delta t)} - \boldsymbol{q}^{(t)} - \dot{\boldsymbol{q}}^{(t)}\Delta t - f_\theta(\boldsymbol{q}^{(t)}, \boldsymbol{h})\Delta t^2||_2, \tag{15}$$

which represents the error for a single timestep. The global truncation error $\mathcal{E}_{t+k\Delta t}$ is defined as:

$$\mathcal{E}_{t+k\Delta t} = ||\boldsymbol{q}^{(t+k\Delta t)} - \boldsymbol{q}_\theta^{(t+k\Delta t)}||_2, \tag{16}$$

which denotes the error accumulated in the first $k$ steps. Then we have

**Corollary 4.4.** *Given the same conditions as in Theorem 4.3, if our learned model adequately approximates the system and $g_\theta$ and $f_\theta$ satisfy the Lipschitz continuity, then, $\forall t \in [t_0, t_1]$ and $k = 1, \cdots, \tau$, the local truncation error $\epsilon_{t+\Delta t}$ and the global truncation error $\mathcal{E}_{t+k\Delta t}$ are the order of $O(\Delta t^2)$ and $O(\Delta t)$, respectively.*

The proof is reported in Appendix A.5. Corollary 4.4 shows that for the prediction of $\tau$-th iteration, its error depends on the chosen $\Delta t$. These statements imply that SEGNO can be trained by minimizing Eq. 12 and generalize to other timesteps.

Though there exist studies (Sanchez-Gonzalez et al., 2020; Bishnoi et al., 2022) that employ accelerations to train models, we remark that SEGNO is different from them and better. As Theorem 4.3 states, we aim to learn the **instantaneous** acceleration of each timestep by minimizing the position discrepancy, while previous works use the **average** acceleration to train the model and then adopt

semi-implicit Euler method to update the next state as well. The average acceleration is computed from the observed trajectory (i.e., $\ddot{q}^{(t_1)} = q^{(t_2)} - 2q^{(t_1)} + q^{(t_0)}$). Thus, we can find that their local truncation error is $O(T)$ even if their loss is zero, while that of SEGNO is $O(\Delta t^2)(T = \tau\Delta t)$. SEGNO theoretically achieves lower error without the need to obtain average acceleration by differentiating future positions.

**Empirical verification** To verify our theoretical findings, we train multiple models of EGNN (Satorras et al., 2021) and SEGNO with EGNN backbone, utilizing different random seeds on a 3-body system with two adjacent states $q^{(t_0)}, q^{(t_1)}$. We derive the intermediate state $q^{(t_{0.5})}$ from their intermediate layers/iterations, which serve as estimations of latent motion trajectories between the two states. In Figure 2, we visualize all predicted trajectories (in dotted grey line), the mean trajectory (in red line), and the mean and variance of predicted states. We can observe that EGNN predictions exhibit high error (red circle) and large variance (blue area) at the intermediate state $q^{(t_{0.5})}$, indicating that the discrete models are incapable of learning the underlying real dynamics from observed system states. Conversely, the learned trajectory of SEGNO demonstrates a significantly lower error and small variance. The additional numerical comparisons are shown in Appendix C.1.

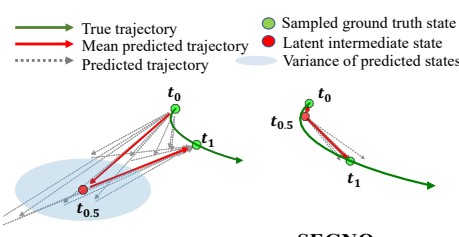

Figure 2: Illustration of learned trajectories from EGNN (left) and SEGNO (right). They are trained to predict the positions of N-body charged systems after 1000ts (See Section 5.1). The green, red, and dotted grey lines are the true, average, and predicted trajectories. $t_0, t_1$ is the observed states. $t_{0.5}$ is the predicted latent state  The blue area denotes the variance.

## 5 EXPERIMENTS

**Datasets** To validate the effectiveness of SEGNO, we first evaluate it on two simulated N-body systems, namely *Charged* particles and *Gravity* particles, which are driven by electromagnetic (Kipf et al., 2018) and gravitational forces (Brandstetter et al., 2021) between each pair of particles, respectively. Subsequently, we compare our model with state-of-the-art models in two challenging datasets: (1) **MD22** (Chmiela et al., 2023) which includes the trajectories of seven large and different types of molecules generated via molecular dynamics simulation; (2) **CMU motion capture** (CMU, 2003), which contains various trajectories of human motions in real-world. Note that all these dataset are symmetric, N-body systems and MD22 are $E(3)$-equivariant, and CMU motion capture is $E(2)$-equivariant[3].

**Baselines** We compare SEGNO with various models: (1) Equivariant models including Radial Field (Köhler et al., 2019), TFN (Thomas et al., 2018), SE(3) Transformer (Fuchs et al., 2020), EGNN (Satorras et al., 2021), GMN (Huang et al., 2022), SEGNN (Brandstetter et al., 2021); (2) Non-equivariant models including GNN, Linear model, and Graph Neural Ordinary Differential Equation (GDE) (Poli et al., 2019).

### 5.1 SIMULATED N-BODY SYSTEM

**Implementation details** We build upon the experimental setting presented in (Satorras et al., 2021) where the task is to estimate all particle positions after a fixed timestep. Each system consists of 5 particles, each with initial positions, velocities, and attributes like positive/negative charge or mass. The graph is fully connected and the initial velocity norm is provided as additional input node features. We employ EGNN (Satorras et al., 2021) as the GNN backbone ($f_\theta$) of SEGNO in these settings. For each system, besides the common setting (Satorras et al., 2021; Brandstetter et al., 2021) with 1000 timesteps (1000 ts), we introduce two extra targets—1500 ts and 2000 ts—which render the learning of latent trajectories more challenging. Other settings including the hyper-parameters are introduced in Appendix B.

---

[3]Symmetry is partially broken by gravity.

Table 1: Mean squared error ($\times 10^{-2}$) of the N-body system. Bold font indicates the best result and underline is the strongest baseline. Results averaged across 5 runs. We report both mean and standard deviation in the table.

| Method | Charged | | | Gravity | | |
| --- | --- | --- | --- | --- | --- | --- |
| | 1000 ts | 1500 ts | 2000 ts | 1000 ts | 1500 ts | 2000 ts |
| Linear | $6.830_{\pm 0.016}$ | $20.012_{\pm 0.029}$ | $39.513_{\pm 0.061}$ | $7.928_{\pm 0.001}$ | $29.270_{\pm 0.003}$ | $58.521_{\pm 0.003}$ |
| GNN | $1.077_{\pm 0.004}$ | $5.059_{\pm 0.250}$ | $10.591_{\pm 0.352}$ | $1.400_{\pm 0.071}$ | $4.691_{\pm 0.288}$ | $10.508_{\pm 0.432}$ |
| GDE | $1.285_{\pm 0.074}$ | $4.026_{\pm 0.164}$ | $8.708_{\pm 0.145}$ | $1.412_{\pm 0.095}$ | $2.793_{\pm 0.083}$ | $6.291_{\pm 0.153}$ |
| TFN | $1.544_{\pm 0.231}$ | $11.116_{\pm 2.825}$ | $23.823_{\pm 3.048}$ | $3.536_{\pm 0.067}$ | $37.705_{\pm 0.298}$ | $73.472_{\pm 0.661}$ |
| SE(3)-Tr. | $2.483_{\pm 0.099}$ | $18.891_{\pm 0.287}$ | $36.730_{\pm 0.381}$ | $4.401_{\pm 0.095}$ | $52.134_{\pm 0.898}$ | $98.243_{\pm 0.647}$ |
| Radial Field | $1.060_{\pm 0.007}$ | $12.514_{\pm 0.089}$ | $26.388_{\pm 0.331}$ | $1.860_{\pm 0.075}$ | $7.021_{\pm 0.150}$ | $16.474_{\pm 0.033}$ |
| EGNN | $0.711_{\pm 0.029}$ | $2.998_{\pm 0.089}$ | $6.836_{\pm 0.093}$ | $0.766_{\pm 0.011}$ | $3.661_{\pm 0.055}$ | $9.039_{\pm 0.216}$ |
| GMN | $0.824_{\pm 0.032}$ | $3.436_{\pm 0.156}$ | $7.409_{\pm 0.214}$ | $0.620_{\pm 0.043}$ | $2.801_{\pm 0.194}$ | $6.756_{\pm 0.427}$ |
| SEGNN | $\underline{0.448}_{\pm 0.003}$ | $\underline{2.573}_{\pm 0.053}$ | $\underline{5.972}_{\pm 0.168}$ | $\underline{0.471}_{\pm 0.026}$ | $\underline{2.110}_{\pm 0.044}$ | $\underline{5.819}_{\pm 0.335}$ |
| SEGNO | $\mathbf{0.433}_{\pm 0.013}$ | $\mathbf{1.858}_{\pm 0.029}$ | $\mathbf{4.285}_{\pm 0.049}$ | $\mathbf{0.338}_{\pm 0.027}$ | $\mathbf{1.362}_{\pm 0.077}$ | $\mathbf{4.017}_{\pm 0.087}$ |

Table 2: Ablation studies ($\times 10^{-2}$) on simulated N-body systems. Results averaged across 5 runs.

| Order | Continuity | Charged | | | Gravity | | |
| --- | --- | --- | --- | --- | --- | --- | --- |
| | | 1000 ts | 1500 ts | 2000 ts | 1000 ts | 1500 ts | 2000 ts |
| First | Discrete | $0.798_{\pm 0.099}$ | $2.215_{\pm 0.058}$ | $4.996_{\pm 0.148}$ | $0.466_{\pm 0.027}$ | $2.342_{\pm 0.424}$ | $5.501_{\pm 0.294}$ |
| Second | Discrete | $0.738_{\pm 0.026}$ | $2.125_{\pm 0.051}$ | $4.948_{\pm 0.198}$ | $0.362_{\pm 0.010}$ | $2.249_{\pm 0.591}$ | $5.343_{\pm 0.263}$ |
| First | Continuous | $\underline{0.537}_{\pm 0.045}$ | $\underline{1.977}_{\pm 0.056}$ | $\underline{4.405}_{\pm 0.159}$ | $\underline{0.352}_{\pm 0.024}$ | $\underline{1.468}_{\pm 0.040}$ | $4.895_{\pm 0.102}$ |
| Second | Continuous | $\mathbf{0.433}_{\pm 0.013}$ | $\mathbf{1.858}_{\pm 0.029}$ | $\mathbf{4.285}_{\pm 0.049}$ | $\mathbf{0.338}_{\pm 0.027}$ | $\mathbf{1.362}_{\pm 0.077}$ | $\mathbf{4.017}_{\pm 0.087}$ |

**Results**   Table 1 depicts the overall results of all models on two datasets. It is evident that SEGNO, equipped solely with EGNN, outperforms all baselines across all datasets and settings. Specifically, compared to the best baseline SEGNN, the average error improvement ($\times 10^{-2}$) on Charged and Gravity datasets is $0.805$ and $0.894$ respectively, demonstrating significant improvement. Additionally, as timesteps increase, the performances of baselines largely drop while SEGNO still can model the latent trajectory. Thus, SEGNO's performance improvement becomes more pronounced. For example, compared to the best baseline SEGNN, the relative improvement in Gravity datasets increases from $28.24\%$ at 1000 time steps to $35.45\%$ at 1500 time steps, demonstrating the strong generalization ability of SEGNO.

### 5.1.1   ABLATION STUDY

In this section, we conduct several ablation experiments on simulated N-body systems to scrutinize our model design, and provide empirical validation for our theoretical findings.

**Physical inductive biases in SEGNO**   To validate the effect of inductive biases incorporated in SEGNO, we construct three variants of SEGNO, each featuring a unique combination of physical inductive biases. Table 2 reports the results where the term 'First' indicates that the model employ $f_\theta$ to parameterize velocity rather than accelerations. 'Discrete' implies that SEGNO does not share the parameters across iterations, akin to discrete models. The original version of SEGNO is denoted as 'Second' and 'Continuous'. From Table 2, we can observe that across all scenarios, continuous models consistently surpass discrete models and the second-order bias consistently enhances the performance of first-order models. These findings serve to corroborate the efficacy of SEGNO's construction and further emphasize the significant advantages of integrating physical inductive biases into the learning process of dynamics.

**Effects of iteration times $\tau$**   We further investigate the impact of the iteration number $\tau$ in SEGNO to empirically verify Corollary 4.4. Given that the target time interval $T$ remains constant, a larger iteration number $\tau$ indicates a smaller interval $\Delta t$. The results are displayed in Figure 3. It is obvious that superior outcomes can be attained by opting for a smaller interval $\Delta t$. Additionally, the performance would not increase after reaching a sufficient number of iterations, which is approximately 10 for both datasets. As per Theorem 4.3, the plateau in performance improvement can be attributed to learning errors which is related to the representative ability of GNN backbone in SEGNO.

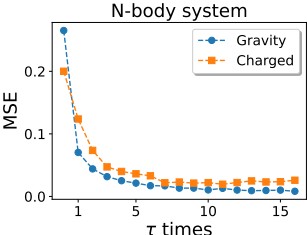

Figure 3: Effects of iteration number $\tau$.

Table 3: Mean squared error ($\times 10^{-3}$) on MD22 dataset. Bold font indicates the best result and Underline is the strongest baseline. Results averaged across 3 runs.

| Molecule | Type | # Atom | TFN | RF | EGNN | GMN | SEGNO |
|---|---|---|---|---|---|---|---|
| Ac-Ala3-NHMe | Protein | 42 | $1.434_{\pm 0.716}$ | $1.340_{\pm 0.072}$ | $\underline{0.539}_{\pm 0.101}$ | $0.960_{\pm 0.157}$ | $\mathbf{0.418}_{\pm 0.038}$ |
| DHA | Lipid | 56 | $0.577_{\pm 0.075}$ | $1.487_{\pm 0.068}$ | $0.782_{\pm 0.035}$ | $\underline{0.453}_{\pm 0.125}$ | $\mathbf{0.370}_{\pm 0.020}$ |
| AT-AT | Nucleic acid | 60 | $1.407_{\pm 0.894}$ | $1.270_{\pm 0.067}$ | $0.583_{\pm 0.053}$ | $\underline{0.568}_{\pm 0.016}$ | $\mathbf{0.440}_{\pm 0.096}$ |
| Stachyose | Carbohydrate | 87 | – | $2.069_{\pm 0.074}$ | $0.629_{\pm 0.073}$ | $\underline{0.583}_{\pm 0.039}$ | $\mathbf{0.548}_{\pm 0.006}$ |
| AT-AT-CG-CG | Nucleic acid | 118 | – | $2.596_{\pm 1.282}$ | $0.609_{\pm 0.031}$ | $\underline{0.581}_{\pm 0.099}$ | $\mathbf{0.394}_{\pm 0.033}$ |
| Buckyball Catcher | Supramolecule | 148 | – | $0.440_{\pm 0.013}$ | $0.554_{\pm 0.093}$ | $\underline{0.309}_{\pm 0.092}$ | $\mathbf{0.199}_{\pm 0.020}$ |
| Double-walled Nanotube | Supramolecule | 370 | – | $0.382_{\pm 0.001}$ | $0.349_{\pm 0.261}$ | $\underline{0.321}_{\pm 0.065}$ | $\mathbf{0.225}_{\pm 0.008}$ |

Table 4: Mean squared error ($\times 10^{-2}$) on CMU motion capture dataset. Bold font indicates the best result and Underline is the strongest baseline. Results averaged across 3 runs.

| Model | TFN | SE(3)-Tr. | RF | EGNN | GMN | SEGNO | Abs. Imp. |
|---|---|---|---|---|---|---|---|
| 30 ts | $24.932_{\pm 1.023}$ | $24.655_{\pm 0.870}$ | $149.459_{\pm 0.750}$ | $24.013_{\pm 0.462}$ | $\underline{16.005}_{\pm 0.386}$ | $\mathbf{14.462}_{\pm 0.106}$ | 1.543 |
| 40 ts | $49.976_{\pm 1.664}$ | $44.279_{\pm 0.355}$ | $306.311_{\pm 1.100}$ | $39.792_{\pm 2.120}$ | $\underline{38.193}_{\pm 0.697}$ | $\mathbf{22.229}_{\pm 1.489}$ | 15.964 |
| 50 ts | $73.716_{\pm 4.343}$ | $68.796_{\pm 1.159}$ | $549.476_{\pm 3.461}$ | $50.930_{\pm 2.675}$ | $\underline{47.883}_{\pm 0.599}$ | $\mathbf{29.264}_{\pm 0.946}$ | 18.619 |

## 5.2 MOLECULAR DYNAMIC

**Implementation details** We use the atomic number and initial velocity norm as input node features. Two atoms are neighbors if their distance is less than a threshold. Our goal is to predict atom positions after 10 data frames. We run experiments on NVIDIA RTX A6000 GPU. TFN and SE(3)-Transformer run out of memory, thus we omit results of SE(3)-Transformer and part of TFN results. We use EGNN (Satorras et al., 2021) as the GNN backbone ($f_\theta$) of SEGNO. Other settings including the hyper-parameters are introduced in Appendix B.

**Results** Table 3 summarizes the results of all models. It is evident that SEGNO outperforms the baselines across 7 molecules, even on the Double-walled Nanotube comprising 370 atoms, supporting the general effectiveness of encoding physical inductive biases. It is worth noting that SEGNO utilizes GMN as its backbone. In comparison to the original GMN, the errors are considerably diminished in all instances. The average relative improvement of SEGNO over GMN on 7 molecules is 15.6%. Such results demonstrate the effectiveness of enhancing physical inductive biases on equivariant GNNs in empirical applications.

## 5.3 CMU MOTION CAPTURE

**Implementation details** CMU Motion Capture (CMU, 2003) contains the trajectories of human motion under several scenarios. Following previous studies (Kipf et al., 2018), we focus on the walking motion of a single object (subject #35). The goal of this task is to predict the data frame after 30 timesteps. Similar to N-body systems, we broaden our assessment scope to include scenarios with intervals of 40 ts and 50 ts, in addition to the default settings with 30 ts. We use GMN as the backbone of SEGNO. The norm of velocity and the coordinates of the gravity axis (z-axis) are set as node features to represent the motion dynamics. Note that the human body operates through joint interactions, two joints are 1-hop neighbors if they are connected naturally and we augment the edges with 2-hop neighbors. Other settings are introduced in Appendix B.

**Results** Table 4 reports the performance of SEGNO and various compared models. We can observe that SEGNO outperforms all baseline models by a significant margin across all scenarios. Notably, the improvements are more pronounced in long-term simulations, with SEGNO achieving $18.619 \times 10^{-2}$ lower MSE than the runner-up model GMN. To gain further insights into the superior performance of SEGNO, we illustrate the predicted motion of GMN and SEGNO in Figure 4.

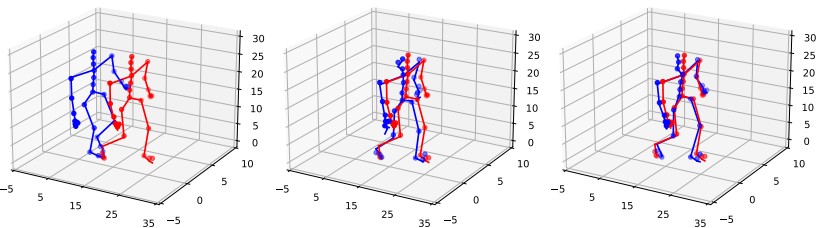

Figure 4: Visualization of Motion Capture with 50 ts. Left to Right: initial position, GMN, SEGNO (all in blue). Ground truths are in red.

Interestingly, it can be observed that the predictions of GMN appear to lag behind the ground truths, while SEGNO demonstrates a closer match. This discrepancy may be attributed to the lack of constraints imposed by modeling latent trajectories. We provide more visualizations in Appendix C.8.

## 6 RELATED WORK

Graph neural networks (Li et al., 2019; Tang et al., 2022; Gao et al., 2023; Tang et al., 2023) have shown promising performance in learning complex physical dynamics. IN (Battaglia et al., 2016), NRI (Kipf et al., 2018), and HRN (Mrowca et al., 2018) are pioneer works that model physical objects and relations as graphs and learn their interaction and evolution. Recently, researchers have taken into account the underlying physical symmetry of systems. TFN (Thomas et al., 2018) and SE(3) Transformer (Fuchs et al., 2020) employ spherical harmonics to construct models with 3D rotation equivariance in the Euclidean group for higher-order geometric representations. LieConv (Finzi et al., 2020a) and LieTransformer (Hutchinson et al., 2021) leverage the Lie convolution to extend equivariance on Lie groups. SEGNN (Brandstetter et al., 2021) proposes to use steerable vectors and their equivariant transformations to represent and process node features. In addition to these methods, a series of studies (Satorras et al., 2021; Huang et al., 2022; Schütt et al., 2021; Wang & Chodera, 2023) apply scalarization techniques to introduce equivariance into the message-passing process in GNNs. Nevertheless, these methods model dynamics in physical systems solely by learning direct mappings between discrete states. They ignore the second-order and continuous nature of observed trajectories, leading to suboptimal generalization performance.

Another research line (Thangamuthu et al., 2022) leverages ODE and Hamiltonian mechanics to capture the interactions in the systems such as Lagrangian Neural Networks (LNN) (Finzi et al., 2020b; Lutter et al., 2019; Bhattoo et al., 2023), Hamiltonian neural networks (HNN) (Greydanus et al., 2019), and Neural ODE (Chen et al., 2018; Gruver et al., 2022; Norcliffe et al., 2020). Recent studies have also enhanced these methods with GNNs. GDE (Poli et al., 2019) and HOGN (Sanchez-Gonzalez et al., 2019) combine GNNs with a differentiable ODE integrator. GNODE (Bishnoi et al., 2022) incorporates a graph-based Neural ODE with additional inductive biases, such as Newton's third law. Compared with these works, we provide theoretical insights that show a second-order Graph Neural ODE can obtain bounded error of instantaneous acceleration and position through minimizing position discrepancy. These findings are further validated in complex applications including molecular and human motion dynamics. In particular, GNS (Sanchez-Gonzalez et al., 2020; Pfaff et al., 2021) also optimizes models via accelerations. However, they only learn the average accelerations that are calculated from the observed trajectories. In contrast, our study focuses on learning instantaneous accelerations and we show it theoretically achieves lower errors than GNS. Furthermore, it is worth mentioning that GNS does not consider equivariance, which is a critical inductive bias that captures symmetries in physical systems. We provide an experimental comparison between them in Appendix C.2.

Besides the above studies, a series of works have combined GNNs and the first-order Neural ODE to learn multi-agent systems (e.g., social networks), including CG-ODE (Huang et al., 2021), LG-ODE (Huang et al., 2020), and HOPE (Luo et al., 2023). However, due to the application difference, they do not consider equivariance and focus on historical state encoding. Thus, they are hard to extend to our task where equivariance and second-order information are vital.

## 7 CONCLUSION

In this work, we address the generalization problem of learning N-body systems and introduce SEGNO, which incorporates the equivariant property from GNN backbones and second-order physical inductive bias. We theoretically prove the uniqueness and boundedness of the trajectories inferred by SEGNO and empirically demonstrate the potential of SEGNO by applying it to a wide range of physical systems. Extensive ablation studies have further substantiated the generalization ability of SEGNO. For future works, we are interested in (1) extending our framework to solve stochastic (Salvi et al., 2022) and partial differential equations (Fortunato et al., 2022; Strönisch et al., 2023; Cao et al., 2023; Xue et al., 2023); (2) jointly considering trajectory forecasting and static tasks such as molecular property (Satorras et al., 2021) or force field predictions (Batzner et al., 2022); (3) integrating more sophisticated physical principles through advanced techniques such as pre-training (Rong et al., 2020; Cheng et al., 2023; Jiao et al., 2023), prompt tuning (Sun et al., 2023) or the utilization of large language models (Li et al., 2023).

ACKNOWLEDGMENTS

The research of Li was supported by NSFC Grant No. 62206067, HKUST-HKUST(GZ) 20 for 20 Cross-campus Collaborative Research Scheme C019 and Guangzhou-HKUST(GZ) Joint Funding Scheme 2023A03J0673. The research of Tsung was supported by NSFC Grant No. 72371271 and Guangzhou Nansha District Key Project 2023ZD003. Particularly, Yu Rong expresses heartfelt gratitude to his wife, Yunman Huang, for the birth of their daughter Xing Rong, which brought immense joy during the pursuit of this research.

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

## A   PROOFS

Note that $h$ is identical for both real and learned systems. Unless otherwise specified, we omit $h$ in the following discussions.

### A.1   PROOF OF PROPOSITION 3.1

According to the constraints of the initial conditions, the initial position $q_\theta^{(t_0)}$ is $E(3)$-equivariant and the initial velocity $\dot{q}_\theta^{(t_0)}$ is $O(3)$-eqivariant and translation-invariant. Given that backbone GNN $f_\theta$ are $O(3)$-equivariant and translation-invariant, and $\mathcal{G}_1$ is $O(3)$-equivariant, for any translation vector $b \in \mathbb{R}^3$, orthogonal matrix $A \in \mathbb{R}^{3\times3}$ and $t' \in [0, \Delta t]$, we have

$$
\begin{aligned}
\Psi_{t',g_\theta,f_\theta}(Aq_\theta^{(t_0)} + b) &= g_\theta(Aq_\theta^{(t_0)} + b) + \mathcal{G}_1(f_\theta(Aq_\theta^{(t_0)} + b), t') \\
&= Ag_\theta(q_\theta^{(t_0)}) + \mathcal{G}_1(Af_\theta(q_\theta^{(t)}), t') \\
&= A\dot{q}_\theta^{(t_0)} + A\mathcal{G}_1(f_\theta(q_\theta^{(t_0)}), t') \\
&= A\dot{q}_\theta^{(t_0+t')} = A\Psi_{\Delta t,g_\theta,f_\theta}(q_\theta^{(t_0)}).
\end{aligned}
\tag{17}
$$

Thus, $\Psi_{t',g_\theta,f_\theta}(q_\theta^{(t_0)})$ is proved to be $O(3)$-equivariant and translation-invariant. Note that $\mathcal{G}_2$ is $O(3)$-equivariant, then, for $q_\theta^{(t_0+t')}$, we have

$$
\begin{aligned}
\Phi_{t',g_\theta}(Aq_\theta^{(t_0)} + b) &= Aq_\theta^{(t_0)} + b + \mathcal{G}_2(\Psi_{t',g_\theta,f_\theta}(Aq_\theta^{(t_0)} + b), t') \\
&= Aq_\theta^{(t_0)} + \mathcal{G}_2(A\Psi_{t',g_\theta,f_\theta}(q_\theta^{(t_0)}), t') + b \\
&= Aq_\theta^{(t_0)} + A\mathcal{G}_2(\Psi_{t',g_\theta,f_\theta}(q_\theta^{(t_0)}), t') + b \\
&= Aq_\theta^{(t_0+t')} + b.
\end{aligned}
\tag{18}
$$

Therefore, $q_\theta^{(t_0+t')} = \Phi_{t',g_\theta}(q_\theta^{(t_0)})$ is $E(3)$-equivariant. Note that the composition of $E(3)$-equivariant functions is still $E(3)$-equivariant. For any time $t \in [t_0, t_1]$, $q_\theta^{(t)}$, which is generated via iteratively calling integrator $\Phi_{t',g_\theta}$ (Eq. 11) with suitable $t' \in [0, \Delta t]$, is $E(3)$-equivariant. Overall, the approximated trajectory $q_\theta$ is $E(3)$-equivariant.

In real-world applications, the assumption of this proposition is generally satisfied. In terms of the equivariance of backbone GNN $f_\theta$, for example, if $f_\theta$ is EGNN whose message passing is defined by

$$
m_{ij}^{(t)} = \phi_e\left(||q_{\theta,i}^{(t)} - q_{\theta,j}^{(t)}||^2, h_i, h_j, a_{ij}\right), \quad \ddot{q}_{\theta,i}^{(t)} = \frac{1}{N-1}\sum_{j \in \mathcal{N}_i}(q_{\theta,i}^{(t)} - q_{\theta,j}^{(t)})\phi_q(m_{ij}^{(t)}).
$$
$$
h_i^{(t+\Delta t)} = h_i^{(t)} + \phi_h(h_i^{(t)}, \sum_{j \in \mathcal{N}_i} m_{ij}^{(t)}).
\tag{19}
$$

Here $\phi_q$ denotes Multi-Layer Perceptron (MLP) whose output is a scalar and the output of $\phi_e, \phi_h$ are vectors. The non-geometric features are updated via skip connections. Analogous to neural ODE methods, the model parameters are shared among all iterations. By (Satorras et al., 2021), it can be easily shown that $f_\theta$ is $O(3)$-equivariant and translation-invariant.

In addition, the increment functions $\mathcal{G}_1, \mathcal{G}_2$ of several widely used numerical integrators for motion dynamics are $O(3)$-equivariant, ensuring that the approximated trajectory is $E(3)$-equivariant. For example, a symplectic Euler integrator computes

$$
q^{(t+\Delta t)} = q^{(t)} + \dot{q}^{(t+\Delta t)}\Delta t, \ \dot{q}^{(t+\Delta t)} = \dot{q}^{(t)} + \ddot{q}^{(t)}\Delta t,
\tag{20}
$$

where $\mathcal{G}_1(x, y) = \mathcal{G}_2(x, y) = x \times y$ is $O(3)$-equivariant. It is straightforward to show that

$$
A\dot{q}^{(t)} + A\ddot{q}^{(t+\Delta t)}\Delta t = A\dot{q}^{(t+\Delta t)}, \ Aq^{(t)} + b + A\dot{q}^{(t+\Delta t)}\Delta t = Aq^{(t+\Delta t)} + b.
\tag{21}
$$

This property also holds for Velocity Verlet

$$
q^{(t+\Delta t)} = q^{(t)} + \dot{q}^{(t)}\Delta t + \frac{1}{2}\ddot{q}^{(t)}\Delta t^2, \ \dot{q}^{(t+\Delta t)} = \dot{q}^{(t)} + \frac{1}{2}(\ddot{q}^{(t)} + \ddot{q}^{(t+\Delta t)})\Delta t,
\tag{22}
$$

and Leapfrog

$$q^{(t+\Delta t)} = q^{(t)} + \dot{q}^{(t)}\Delta t + \frac{1}{2}\ddot{q}^{(t)}\Delta t^2, \ \dot{q}^{(t+\Delta t)} = \dot{q}^{(t)} + \ddot{q}^{(t+\Delta t)}\Delta t. \tag{23}$$

The proof is the same as symplectic Euler.

## A.2 Proof of Lemma 4.1

We first prove the uniqueness of the solution in our dynamic system. The key to this proof is to convert the high-order non-linear ODE to a first-order non-linear ODE. Let's define a new variable $Q^{(t)} = (\dot{q}^{(t)}, q^{(t)})$ and $\dot{Q}^{(t)}$ as its first-order derivative with respect $t$. Then in terms of this new variable, the second-order non-linear ODE becomes

$$\dot{Q}^{(t)} = (F(Q^{(t)}), G(Q^{(t)})), \tag{24}$$

where $F(Q^{(t)})) = f(q^{(t)})$ and $G(Q^{(t)}) = \dot{q}^{(t)}$. Furthermore, let's define $H(Q^{(t)}) = (F(Q^{(t)}), G(Q^{(t)}))$, and Eq. 24 is reformatted as

$$\dot{Q}^{(t)} = H(Q^{(t)}), \quad Q^{(t_0)} = (\dot{q}^{(t_0)}, q^{(t_0)}), \tag{25}$$

which is exactly a first-order non-linear ODE with a known initial condition. Note that $G$ and $f$ are both Lipschitz continuous, thus $H$ is Lipschitz continuous. Then, based on Picard's existence theorem (Coddington et al., 1956), the aforementioned non-linear ODE has a unique solution $Q^{(t)}$, $\forall t \in [t_0, t_1]$. Subsequently, our system has a unique solution $q^{(t)}$, $\forall t \in [t_0, t_1]$.

## A.3 Proof of Proposition 4.2

Due to the uniqueness of the solution, if the realistic measurement $q^{(t_1)}$ is given, then the trajectory $q^{(t)}$ is fully determined over the time interval $[t_0, t_1]$. Under SEGNO framework, we define the discrepancy between $q_\theta^{(t_1)}$ and $q^{(t_1)}$ as

$$d(q_\theta^{(t_1)}, q^{(t_1)}) = ||q_\theta^{(t_1)} - q^{(t_1)}||_p = ||(\Phi_{\Delta t, g_\theta})^\tau(q^{(t_0)}) - \phi_{T,g}(q^{(t_0)})||_p, \tag{26}$$

where $|| \cdot ||_p$ represents the $p$-norm. According to Eq. 9, $\Phi_{\Delta t, g_\theta}$ is fully determined by $f_\theta$. Without loss of generality, we take SEGNO with Euler integrators as an example. Given the known initial position and velocity, along with neural ODE update scheme in SEGNO, we can convert Eq. 26 to the following form

$$d(\mathbf{q}_\theta^{(t_1)}, \mathbf{q}^{(t_1)}) = ||\mathbf{q}_\theta^{(t_1)} - \mathbf{q}^{(t_1)}||_p$$
$$= ||\sum_{k=0}^{\tau-1}(\int_{t_0+k\Delta t}^{t_0+(k+1)\Delta t}(\dot{\mathbf{q}}^{(t_0+k\Delta t)} + \int_{t_0+k\Delta t}^{t} f(\mathbf{q}^{(m)})dm)dt - (\dot{\mathbf{q}}_\theta^{(t_0+k\Delta t)}\Delta t + f_\theta(\mathbf{q}_\theta^{(t_0+k\Delta t)})\Delta t^2)||_p$$
$$= ||\sum_{k=0}^{\tau-1}\mathcal{D}(\dot{\mathbf{q}}^{(t_0+k\Delta t)}, f(\mathbf{q}^{(t_0+k\Delta t)}), \dot{\mathbf{q}}_\theta^{(t_0+k\Delta t)}, f_\theta(\mathbf{q}_\theta^{(t_0+k\Delta t)}))||_p, \tag{27}$$

where $\mathcal{D}(\cdot)$ denotes the discrepancy function. Considering that there are no sudden changes in velocity or acceleration for a smooth trajectory, with sufficiently small $\Delta t$, we can approximate $\mathcal{D}(\cdot)$ as follows

$$\mathcal{D}(\dot{\mathbf{q}}^{(t_0+k\Delta t)}, f(\mathbf{q}^{(t_0+k\Delta t)}), \dot{\mathbf{q}}_\theta^{(t_0+k\Delta t)}, f_\theta(\mathbf{q}_\theta^{(t_0+k\Delta t)}))$$
$$\approx (\dot{\mathbf{q}}^{(t_0+k\Delta t)} - \dot{\mathbf{q}}_\theta^{(t_0+k\Delta t)})\Delta t + (f(\mathbf{q}^{(t_0+k\Delta t)}) - f_\theta(\mathbf{q}_\theta^{(t_0+k\Delta t)})\Delta t^2$$
$$\approx \sum_{i=0}^{k}(f(\mathbf{q}^{(t_0+i\Delta t)}) - f_\theta(\mathbf{q}_\theta^{(t_0+i\Delta t)})\Delta t^2. \tag{28}$$

Note that Lemma 4.1 guarantees the uniqueness of target trajectory as well as its acceleration. Therefore, according to the aforementioned analysis, SEGNO tends to let $f_\theta(\mathbf{q}_\theta^{(t_0+k\Delta t)})$ approximate the unique accelaration $f(\mathbf{q}^{(t_0+k\Delta t)})$ via minimizing position loss. Then, under the assumption of widely used universal approximation theorem (Hornik et al., 1989), there exists a $f_{\theta^*}$ such that $f_{\theta^*}(q_{\theta^*}^{(t)}, h) = f(q^{(t)}, h), \forall t \in [t_0, t_1]$.

## A.4 PROOF OF THEOREM 4.3

The proof sketch is as follows: We first revisit the core definitions pertaining to neural ODE in SEGNO and introduce its variant with Euler integrator, then derive the bound for first-order approximation error $||g_\theta - g||_\infty$, and finally extend the results to the second-order case $||f_\theta - f||_\infty$ to finish the proof.

### A.4.1 EULER INTEGRATOR IN SEGNO

As mentioned in Eq. 10, the Euler integrator $\Phi_{\Delta t, g_\theta}$, as mentioned in Eq. 20, approaches $\phi_{\Delta t, g_\theta}$ via

$$
\begin{aligned}
g_\theta(\boldsymbol{q}_\theta^{(t+\Delta t)}) &= g_\theta(\boldsymbol{q}_\theta^{(t)}) + f_\theta(\boldsymbol{q}_\theta^{(t)})\Delta t, \\
\Phi_{\Delta t, g_\theta}(\boldsymbol{q}_\theta^{(t)}) &= \boldsymbol{q}_\theta^{(t)} + g_\theta(\boldsymbol{q}_\theta^{(t+\Delta t)})\Delta t.
\end{aligned}
\tag{29}
$$

Considering Eq. 11 and 12, it is straightforward to reframe our learning objective in the context of a neural ODE:

$$
\mathcal{L}_{\text{train}} = \sum_{s \in \mathcal{D}_{\text{train}}} ||\boldsymbol{q}_{\theta,s}^{(t_1)} - \boldsymbol{q}_s^{(t_1)}||^2 = \sum_{s \in \mathcal{D}_{\text{train}}} ||(\Phi_{\Delta t, g_\theta})^\tau (\boldsymbol{q}_s^{(t_0)}) - \phi_{T,g}(\boldsymbol{q}_s^{(t_0)})||^2.
\tag{30}
$$

### A.4.2 APPROXIMATION ERROR OF $f_\theta$

In our dynamical system, $g_\theta$ and $g$ are entirely determined by $f_\theta$ and $f$ respectively. Thus, we first establish the boundedness of $||g_\theta - g||_\infty$, then demonstrate the approximation error of $||f_\theta - f||_\infty$.

**Lemma A.1.** *For $\boldsymbol{q}^{(t_0)} \in \mathbb{R}^3$ as the initial position of a trajectory, $r_a, r_b, T, \tau > 0$ and $k \in \mathbb{Z}^+$, a given ODE solver that is $\tau$ compositions of an Euler Integrator $\Phi_{\Delta t, g_\theta}$ with $\Delta t = T/\tau$, a set of points on exact trajectory associated with time increment interval $[k\Delta t, T + k\Delta t]$ as*

$$
V_{k\Delta t}^{T+k\Delta t} = \{\phi_{t',g}(\boldsymbol{q}^{(t_0)}) | k\Delta t \le t' \le T + k\Delta t\},
\tag{31}
$$

*, and denote*

$$
\mathcal{L}_{k\Delta t}^{T+k\Delta t} = \frac{1}{T} ||(\Phi_{\Delta t, g_\theta})^\tau - \phi_{T,g}||_{\mathcal{B}(V_{k\Delta t}^{T+k\Delta t}, r_a)},
\tag{32}
$$

*and suppose that $g$ and $g_\theta$ are analytic and bounded by $m$ within $\mathcal{B}(V_{k\Delta t}^{T+k\Delta t}, r_a + r_b)$, the union of complex balls centered at $\boldsymbol{q} \in V_{k\Delta t}^{T+k\Delta t}$ with radius $r_a + r_b$. Then, there exist constants $T_0$ and $C$ that depends on $r_a/m$, $r_b/m$, $\tau$, and $\Phi_{\Delta t, g_\theta}$, such that, if $0 < T < T_0$, $\forall t \in [t_0 + k\Delta t, t_0 + T + k\Delta t]$,*

$$
||g_\theta(\boldsymbol{q}^{(t)}) - g(\boldsymbol{q}^{(t)})||_\infty \le Cm\Delta t + \frac{e}{e-1} \mathcal{L}_{k\Delta t}^{T+k\Delta t},
\tag{33}
$$

*where $e$ is the base of the natural logarithm.*

*Proof.* This result can be directly derived from Theorem 3.1, 3.2 and Corollary 3.3 in (Zhu et al., 2022) via replacing the integrator from Runge-Kutta integrator with Euler integrator since Euler integrator has been proven to satisfy the prerequisites of theorems in Appendix B.2 in (Zhu et al., 2022). □

Since $\mathcal{B}(V_0^T, r_1) \subset \mathcal{B}(V_0^{2T}, r_1)$, per our assumption, then $g$ and $g_\theta$ are both analytic and bounded by $m_1$ in $\mathcal{B}(V_0^T, r_1)$. To utlize Lemma A.1, we set $k = 0$ and $T = t_1 - t_0$. With suitable $r_1$ and $m_1$, we have $T < T_0$ and, $\forall t \in [t_0, t_1]$,

$$
||g_\theta(\boldsymbol{q}^{(t)}) - g(\boldsymbol{q}^{(t)})||_\infty \le C_1 m_1 \Delta t + \frac{e}{e-1} \mathcal{L}_0^T,
\tag{34}
$$

where $\mathcal{L}_0^T = \frac{1}{T} ||(\Phi_{\Delta t, g_\theta})^\tau - \phi_{T,g}||_{\mathcal{B}(V_0^T, r_1)}$ and $C_1$ is a control constant.

To establish the connection between $g$ and $f$, we only focus on the first time step $\Delta t$ instead of the entire time interval $T$. Recall that $g$ is obtained by integrating $f$, we have

$$
\psi_{\Delta t, g, f}(\boldsymbol{q}^{(t_0)}) = g(\boldsymbol{q}^{(t_0 + \Delta t)}) = g(\boldsymbol{q}^{(t_0)}) + \int_{t_0}^{t_0 + \Delta t} f(\boldsymbol{q}^{(t)}) \, dt,
\tag{35}
$$

as the flow map for the first-order derivative of the exact trajectory in a single step. As defined in Eq. 29, its corresponding Euler integrator has the form

$$g_\theta(\boldsymbol{q}_\theta^{(t_0+\Delta t)}) = \Psi_{\Delta t, g_\theta, f_\theta}(\boldsymbol{q}^{(t_0)}) = g_\theta(\boldsymbol{q}_\theta^{(t_0)}) + f_\theta(\boldsymbol{q}_\theta^{(t_0)})\Delta t. \tag{36}$$

Note that $\mathcal{B}(V_0^T, r_2) \subset \mathcal{B}(V_0^{2T}, r_2)$, $f$ and $f_\theta$ are both analytic and bounded by $m_2$ in $\mathcal{B}(V_0^T, r_2)$. In lemma A.1, we substitute $\phi_{\Delta t, g_\theta}$ by $\Psi_{\Delta t, g_\theta, f_\theta}$ and set $T = \Delta t$ with $\tau = 1$ and $k = 0$, then the corresponding loss becomes

$$\begin{aligned}
\mathcal{L}_{0,1}^T &= \frac{1}{\Delta t}||\Psi_{\Delta t, g_\theta, f_\theta} - \psi_{\Delta t, g, f}||_{\mathcal{B}(V_0^{\Delta t}, r_2)} \\
&= \sup_{\boldsymbol{q} \in \mathcal{B}(V_0^{\Delta t}, r_2)} \frac{1}{\Delta t}||\Psi_{\Delta t, g_\theta, f_\theta}(\boldsymbol{q}) - \psi_{\Delta t, g, f}(\boldsymbol{q})||_\infty \\
&= \frac{1}{\Delta t}||\Psi_{\Delta t, g_\theta, f_\theta}(\boldsymbol{q}^*) - \psi_{\Delta t, g, f}(\boldsymbol{q}^*)||_\infty \\
&= \frac{1}{\Delta t}||g_\theta(\tilde{\boldsymbol{q}}_\theta^{(t_0+\Delta t)}) - g(\tilde{\boldsymbol{q}}^{(t_0+\Delta t)})||_\infty,
\end{aligned} \tag{37}$$

where $\boldsymbol{q}^*$ denote the point that maximizes the Eq. 37 and $\tilde{\boldsymbol{q}}^{(t)}$ is another trajectory with $\tilde{\boldsymbol{q}}^{(t_0)} = \boldsymbol{q}^*$ because $\boldsymbol{q}^* \in V_0^{\Delta t}$ may not holds. For both $\tilde{\boldsymbol{q}}^{(t)}$ and $\boldsymbol{q}^t$, the actual position and estimation from SEGNO at $t_0 + \Delta t$ have the form

$$\begin{aligned}
\tilde{\boldsymbol{q}}^{(t_0+\Delta t)} &= \tilde{\boldsymbol{q}}^{(t_0)} + \int_{t_0}^{t_0+\Delta t} g(\tilde{\boldsymbol{q}}^{(t)})\, dt, \\
\tilde{\boldsymbol{q}}_\theta^{(t_0+\Delta t)} &= \tilde{\boldsymbol{q}}^{(t_0)} + g_\theta(\tilde{\boldsymbol{q}}^{(t_0)})\Delta t + f_\theta(\tilde{\boldsymbol{q}}^{(t_0)})\Delta t^2, \\
\boldsymbol{q}_\theta^{(t_0+\Delta t)} &= \boldsymbol{q}^{(t_0)} + g_\theta(\boldsymbol{q}^{(t_0)})\Delta t + f_\theta(\boldsymbol{q}^{(t_0)})\Delta t^2.
\end{aligned} \tag{38}$$

Then, with suitable $\Delta t$ and $r_1$, $g(\tilde{\boldsymbol{q}}^{(t_0+\Delta t)})$, $g_\theta(\tilde{\boldsymbol{q}}_\theta^{(t_0+\Delta t)})$ and $g_\theta(\boldsymbol{q}_\theta^{(t_0+\Delta t)})$ are bounded by $m_1$ since $\tilde{\boldsymbol{q}}^{(t_0+\Delta t)}, \tilde{\boldsymbol{q}}_\theta^{(t_0+\Delta t)}, \boldsymbol{q}_\theta^{(t_0+\Delta t)} \in \mathcal{B}(V_T, r_1)$. Given such, the aforementioned loss is transformed into

$$\begin{aligned}
\mathcal{L}_{0,1}^T &\le \frac{1}{\Delta t}\big[||g_\theta(\tilde{\boldsymbol{q}}_\theta^{(t_0+\Delta t)}) - g_\theta(\tilde{\boldsymbol{q}}^{(t_0+\Delta t)})||_\infty + ||g_\theta(\tilde{\boldsymbol{q}}^{(t_0+\Delta t)}) - g_\theta(\boldsymbol{q}^{(t_0+\Delta t)})||_\infty \\
&\quad + ||g_\theta(\boldsymbol{q}^{(t_0+\Delta t)}) - g(\boldsymbol{q}^{(t_0+\Delta t)})||_\infty + ||g(\boldsymbol{q}^{(t_0+\Delta t)}) - g(\tilde{\boldsymbol{q}}^{(t_0+\Delta t)})||_\infty\big] \\
&\le \frac{1}{\Delta t}(6m_1 + C_1 m_1 \Delta t + \frac{e}{e-1}\mathcal{L}_0^T).
\end{aligned} \tag{39}$$

Subsequently, given that $\Delta t < T < T_0$, we have, $\forall t \in [t_0, t_0 + \Delta t]$,

$$\begin{aligned}
||f_\theta(\boldsymbol{q}^{(t)}) - f(\boldsymbol{q}^{(t)})||_\infty &\le C_2 m_2 \Delta t + \frac{e}{e-1}\mathcal{L}_{0,1}^T, \\
&\le C_2 m_2 \Delta t + \frac{e}{e-1}\big[\frac{1}{\Delta t}(6m_1 + C_1 m_1 \Delta t + \frac{e}{e-1}\mathcal{L}_0^T)\big], \\
&\le C_2 m_2 \Delta t + (\frac{e}{e-1})^2 \frac{\mathcal{L}_0^T}{\Delta t} + \frac{e}{e-1}(\frac{6m_1}{\Delta t} + C_1 m_1), \\
&\le O(\Delta t + \frac{\mathcal{L}_0^T}{\Delta t}).
\end{aligned} \tag{40}$$

To extend the boundness up to $t_1$, we can easily utilize Lemma A.1 with different $k = 1, \cdots, \tau - 1$ (Eq. 34) to repeat the above derivation, and obtain, $\forall t \in [t_0 + k\Delta t, t_0 + (k+1)\Delta t]$,

$$||f_\theta(\boldsymbol{q}^{(t)}) - f(\boldsymbol{q}^{(t)})||_\infty \le O(\Delta t + \frac{\mathcal{L}_{k\Delta t}^{T+k\Delta t}}{\Delta t}), \tag{41}$$

where $\mathcal{L}_{k\Delta t}^{T+k\Delta t} = \frac{1}{T}||(\Phi_{\Delta t, g_\theta})^\tau - \phi_{T,g}||_{\mathcal{B}(V_{k\Delta t}^{T+k\Delta t}, r_1)}$. Therefore, $\forall t \in [t_0, t_1]$,

$$\begin{aligned}
||f_\theta(\boldsymbol{q}^{(t)}) - f(\boldsymbol{q}^{(t)})||_\infty &\le \sup_{k=0,\cdots,\tau-1} O(\Delta t + \frac{\mathcal{L}_{k\Delta t}^{T+k\Delta t}}{\Delta t}) \\
&\le O(\Delta t + \frac{\mathcal{L}_0^{2T}}{\Delta t}),
\end{aligned} \tag{42}$$

where $\mathcal{L}_0^{2T} = \frac{1}{T}||(\Phi_{\Delta t, g_\theta})^\tau - \phi_{T,g}||_{\mathcal{B}(V_0^{2T}, r_1)}$ and it concludes the proof.

## A.5 PROOF OF COROLLARY 4.4

### A.5.1 LOCAL TRUNCATION ERROR

We use Taylor series expansion to approximate the trajectory at time $t + \Delta t$ as

$$\boldsymbol{q}^{(t+\Delta t)} = \boldsymbol{q}^{(t)} + g(\boldsymbol{q}^{(t)})\Delta t + \frac{1}{2}f(\boldsymbol{q}^{(t)})\Delta t^2 + O(\Delta t^3). \tag{43}$$

Then the local truncation error $\epsilon_{t+\Delta t}$, $\forall t \in [t_0, t_1]$, equals to

$$
\begin{aligned}
\epsilon_{t+\Delta t} &= ||\boldsymbol{q}^{(t+\Delta t)} - \boldsymbol{q}^{(t)} - g(\boldsymbol{q}^{(t)})\Delta t - f_\theta(\boldsymbol{q}^{(t)})\Delta t^2||_2 \\
&= ||\boldsymbol{q}^{(t)} + g(\boldsymbol{q}^{(t)})\Delta t + \frac{1}{2}f(\boldsymbol{q}^{(t)})\Delta t^2 + O(\Delta t^3) \\
&\quad - \boldsymbol{q}^{(t)} - g(\boldsymbol{q}^{(t)})\Delta t - f_\theta(\boldsymbol{q}^{(t)})\Delta t^2||_2 \\
&= ||\frac{1}{2}f(\boldsymbol{q}^{(t)})\Delta t^2 - f_\theta(\boldsymbol{q}^{(t)})\Delta t^2 + O(\Delta t^3)||_2 \\
&= ||\frac{1}{2}(f(\boldsymbol{q}^{(t)}) - f_\theta(\boldsymbol{q}^{(t)}))\Delta t^2 - \frac{1}{2}f_\theta(\boldsymbol{q}^{(t)})\Delta t^2 + O(\Delta t^3)||_2 \\
&\leq ||\frac{1}{2}(f(\boldsymbol{q}^{(t)}) - f_\theta(\boldsymbol{q}^{(t)}))\Delta t^2||_2 + ||\frac{1}{2}f_\theta(\boldsymbol{q}^{(t)})\Delta t^2||_2 + O(\Delta t^3) \\
&\leq \frac{\sqrt{3}\Delta t^2}{2}||f(\boldsymbol{q}^{(t)}) - f_\theta(\boldsymbol{q}^{(t)}))||_\infty + ||\frac{1}{2}m_2\Delta t^2||_2 + O(\Delta t^3) \\
&\leq \frac{\sqrt{3}\Delta t^2}{2}O(\Delta t + \frac{2\mathcal{L}_0^{2T}}{\Delta t}) + O(\Delta t^2).
\end{aligned}
\tag{44}
$$

Here the last inequality is due to Theorem 4.3. If the learned model achieves a near-perfect approximation of the true trajectory, resulting in a significantly diminished loss $\mathcal{L}_0^{2T}$. Then, $\forall t \in [t_0, t_1]$,

$$\epsilon_{t+\Delta t} \leq O(\Delta t^2). \tag{45}$$

### A.5.2 GLOBAL TRUNCATION ERROR

Considering $t \in [t_0, t_1]$ and $k = 1, \cdots, \tau - 1$, the boundness of the global truncation error can be derived in a recursive way via

$$
\begin{aligned}
\mathcal{E}_{t+(k+1)\Delta t} &= ||\boldsymbol{q}^{(t+(k+1)\Delta t)} - \boldsymbol{q}_\theta^{(t+(k+1)\Delta t)}||_2 \\
&= ||\boldsymbol{q}^{(t+(k+1)\Delta t)} - \boldsymbol{q}_\theta^{(t+k\Delta t)} - g_\theta(\boldsymbol{q}_\theta^{(t+k\Delta t)})\Delta t - f_\theta(\boldsymbol{q}_\theta^{(t+k\Delta t)})\Delta t^2||_2 \\
&= ||\boldsymbol{q}^{(t+k\Delta t)} - \boldsymbol{q}_\theta^{(t+k\Delta t)} + \boldsymbol{q}^{(t+(k+1)\Delta t)} - \boldsymbol{q}^{(t+k\Delta t)} - g(\boldsymbol{q}^{(t+k\Delta t)})\Delta t \\
&\quad - f_\theta(\boldsymbol{q}^{(t+k\Delta t)})\Delta t^2 + f_\theta(\boldsymbol{q}^{(t+k\Delta t)})\Delta t^2 - f_\theta(\boldsymbol{q}_\theta^{(t+k\Delta t)})\Delta t^2 \\
&\quad + g(\boldsymbol{q}^{(t+k\Delta t)})\Delta t - g_\theta(\boldsymbol{q}^{(t+k\Delta t)})\Delta t \\
&\quad + g_\theta(\boldsymbol{q}^{(t+k\Delta t)})\Delta t - g_\theta(\boldsymbol{q}_\theta^{(t+k\Delta t)})\Delta t||_2 \\
&\leq \mathcal{E}_{t+k\Delta t} + \epsilon_{t+(k+1)\Delta t} + ||f_\theta(\boldsymbol{q}^{(t+k\Delta t)}) - f_\theta(\boldsymbol{q}_\theta^{(t+k\Delta t)})||_2\Delta t^2 \\
&\quad + \left[||g_\theta(\boldsymbol{q}^{(t+k\Delta t)}) - g_\theta(\boldsymbol{q}_\theta^{(t+k\Delta t)})||_2 + ||g(\boldsymbol{q}^{(t+k\Delta t)}) - g_\theta(\boldsymbol{q}^{(t+k\Delta t)})||_2\right]\Delta t.
\end{aligned}
\tag{46}
$$

Note that $g_\theta$ and $f_\theta$ satisfy the Lipschitz continuity, we have

$$
\begin{aligned}
||g_\theta(\boldsymbol{q}^{(t+k\Delta t)}) - g_\theta(\boldsymbol{q}_\theta^{(t+k\Delta t)})||_2 &\leq L_g||\boldsymbol{q}^{(t+k\Delta t)} - \boldsymbol{q}_\theta^{(t+k\Delta t)}||_2 = L_g||\mathcal{E}_{t+k\Delta t}||_2, \\
||f_\theta(\boldsymbol{q}^{(t+k\Delta t)}) - f_\theta(\boldsymbol{q}_\theta^{(t+k\Delta t)})||_2 &\leq L_f||\boldsymbol{q}^{(t+k\Delta t)} - \boldsymbol{q}_\theta^{(t+k\Delta t)}||_2 = L_f||\mathcal{E}_{t+k\Delta t}||_2,
\end{aligned}
\tag{47}
$$

where $L_g$ and $L_f$ denote Lipschitz constant for $g_\theta$ and $f_\theta$ respectively. Given that the learned model achieves a near-perfect approximation of the true trajectory, by Lemma A.1 and Eq. 34, we obtain

$$||g(\boldsymbol{q}^{(t+k\Delta t)}) - g_\theta(\boldsymbol{q}^{(t+k\Delta t)})||_2 \leq \sqrt{3}\left[C_1 m_1 \Delta t + \frac{e}{e-1}\mathcal{L}_0^{2T}\right] \leq O(\Delta t). \tag{48}$$

Thus, the global truncation error in Eq. 46 is transformed into

$$\mathcal{E}_{t+(k+1)\Delta t} \leq (1 + L_g\Delta t + L_f\Delta t^2)\mathcal{E}_{t+k\Delta t} + O(\Delta t^2). \tag{49}$$

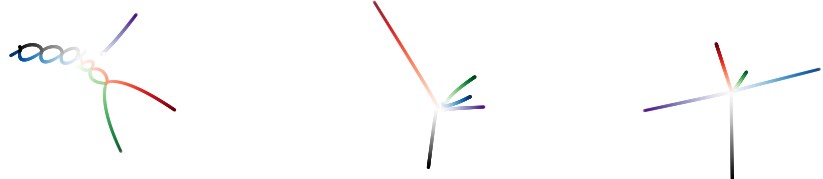

Figure 5: Example trajectories of 5-body charged system. From left to right, the number of positive and negative charges are 1, 3, 0.

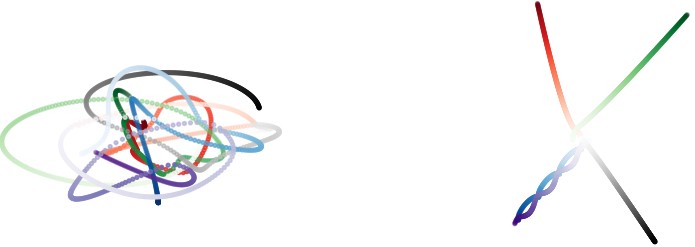

Figure 6: Example trajectories of 5-body gravity system. The system formed by gravitational force can either aggregate (i.e., left) or disperse (i.e., right).

Note that the global truncation error for the first step $\mathcal{E}_{t+\Delta t}$, where $k = 1$, is exactly the local truncation error $\epsilon_{t+\Delta t}$ mentioned in Eq. 44. Given that $k \leq \tau$, we can derive, $\forall t \in [t_0, t_1]$ and $k = 1, \cdots, \tau$,

$$
\begin{aligned}
\mathcal{E}_{t+k\Delta t} &\leq (1 + L_g\Delta t + L_f\Delta t^2)^{k-1}O(\Delta t^2) + \cdots + (1 + L_g\Delta t + L_f\Delta t^2)O(\Delta t^2) \\
&\leq O(\tau\Delta t^2) = O(\frac{T}{\Delta t}\Delta t^2) = O(\Delta t).
\end{aligned}
\tag{50}
$$

## B  IMPLEMENTATION DETAILS

### B.1  MORE DETAILS ON SIMULATED N-BODY SYSTEMS

**N-body charged system**  We use the same N-body charged system code[4] with previous work (Satorras et al., 2021; Brandstetter et al., 2021). They inherit the 2D implementation of (Kipf et al., 2018) and extend it to 3 dimensions. System trajectories are generated in 0.001 timestep and unbounded with virtual boxes. The initial location is sampled from a Gaussian distribution (mean $\mu = 0$, standard deviation $\sigma = 0.5$), and the initial velocity is a random vector of norm 0.5. According to the charge types, three types of systems exist where the difference between the number of positive and negative charges are 1, 3, and 0. Example trajectories of these three types of systems are provided in Figure 5.

**N-body gravity system**  The code[5] of gravitational N-body systems is provided by (Brandstetter et al., 2021). They implement it under the same framework as the above charged N-body systems. System trajectories are generated in 0.001 timestep, using gravitational interaction and no boundary conditions. Particle positions are initialized from a unit Gaussian, particle velocities are initialized with a norm equal to one, random direction, and particle mass is set to one. The system formed

---

[4]https://github.com/vgsatorras/egnn
[5]https://github.com/RobDHess/Steerable-E3-GNN

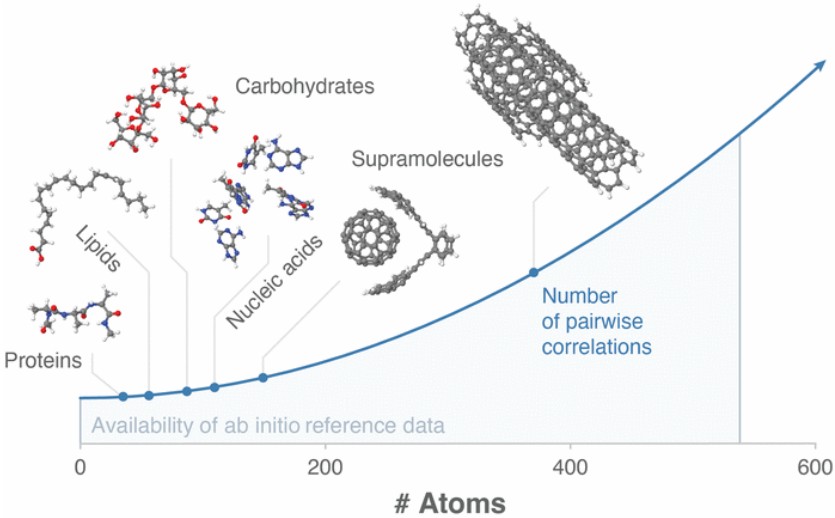

Figure 7: Molecular structures of MD22 dataset, which is borrowed from Fig.1 of original paper (Chmiela et al., 2023).

by gravitational force can either aggregate or disperse. Example trajectories of these two types of systems are provided in Figure 6.

**Hyperparameter** We empirically find that the following hyperparameters generally work well, and use them across most experimental evaluations: Adam optimizer with learning rate 0.001, the number of epochs 500, hidden dim 64, weight decay $1 \times 10^{-12}$, and layer number 4. We set the iteration time of SEGNO to 8. The representation degrees of SE(3)-transformer and TFN are set to 3 and 2. The number of training, validation, and testing sets are 3000, 2000, and 2000, respectively.

### B.2 MORE DETAILS ON MD22

**Molecules and Hyperparameter** The molecular structures of MD22 are displayed in Figure 7, which is borrowed from their paper (Chmiela et al., 2023). We use the following hyperparameters across all experimental evaluations: Adam optimizer with learning rate 0.001, the number of epochs 5000 with early stopping 100, hidden dim 64, weight decay $1 \times 10^{-12}$, and layer number 4. The iteration time of SEGNO is searched from 4 to 6. The representation degrees of SE(3)-transformer and TFN are set to 2. Due to the limited memory, the batch size of TFN is set to 10. The number of training, validation, and testing sets are 500, 2000, and 2000, respectively. The threshold for graph construction is set to 2.5 for all molecules.

### B.3 MORE DETAILS ON MOTION CAPTURE

**Hyperparameter** We use the following hyperparameters across all experimental evaluations: Adam optimizer with learning rate 0.001, the number of epochs 3000, hidden dim 64, weight decay $1 \times 10^{-12}$, and layer number 4. The iteration time of SEGNO is set to 4. We adopt a random split strategy introduced by Huang et al. (2022) where train/validation/test data contains 200/600/600 frame pairs.

## C ADDITIONAL EXPERIMENTS

### C.1 ACCURACY OF LEARNED LATENT TRAJECTORY

It is interesting to see how models learn the latent trajectory between the input and output states. Accordingly, we train models on 1000ts on two datasets and make the test on shorter time steps

Table 5: The generalization from long-term to short-term. All models are trained on 1000ts and test on 250/500/750/1000 ts. Mean squared error ($\times 10^{-2}$) and the standard deviation are reported. Results averaged across 5 runs.

| Method | Charged | | | | | Gravity | | | | |
|---|---|---|---|---|---|---|---|---|---|---|
| | 250 ts | 500 ts | 750 ts | 1000 ts | Avg | 250 ts | 500 ts | 750 ts | 1000 ts | Avg |
| GNN | $73.40_{\pm 9.60}$ | $31.79_{\pm 5.28}$ | $12.86_{\pm 2.81}$ | $0.826_{\pm 0.08}$ | 29.72 | $181.9_{\pm 26.1}$ | $90.33_{\pm 15.5}$ | $30.66_{\pm 12.3}$ | $0.746_{\pm 0.05}$ | 75.93 |
| GDE | $92.65_{\pm 25.0}$ | $43.94_{\pm 10.9}$ | $12.20_{\pm 23.2}$ | $0.652_{\pm 0.05}$ | 37.36 | $136.0_{\pm 135}$ | $56.80_{\pm 60.6}$ | $12.21_{\pm 14.8}$ | $0.588_{\pm 0.58}$ | 51.39 |
| EGNN | $6.756_{\pm 3.05}$ | $3.816_{\pm 4.68}$ | $3.668_{\pm 0.74}$ | $0.568_{\pm 0.09}$ | 3.702 | $7.146_{\pm 7.06}$ | $29.70_{\pm 20.4}$ | $9.712_{\pm 3.60}$ | $0.382_{\pm 0.11}$ | 11.89 |
| GMN | $10.44_{\pm 7.43}$ | $10.92_{\pm 4.67}$ | $4.518_{\pm 1.36}$ | $0.512_{\pm 0.16}$ | 6.598 | $7.430_{\pm 8.19}$ | $9.540_{\pm 12.1}$ | $5.730_{\pm 6.69}$ | $0.349_{\pm 0.48}$ | 5.762 |
| SEGNN | $21.78_{\pm 9.69}$ | $52.74_{\pm 15.6}$ | $34.13_{\pm 14.9}$ | $0.342_{\pm 0.04}$ | 27.25 | $10.58_{\pm 6.28}$ | $49.63_{\pm 37.7}$ | $25.82_{\pm 27.0}$ | $0.448_{\pm 0.02}$ | 21.62 |
| SEGNO | $0.188_{\pm 0.03}$ | $0.312_{\pm 0.06}$ | $0.360_{\pm 0.06}$ | $0.309_{\pm 0.11}$ | 0.292 | $0.064_{\pm 0.02}$ | $0.128_{\pm 0.03}$ | $0.176_{\pm 0.04}$ | $0.210_{\pm 0.07}$ | 0.145 |

by performing SEGNO on the smaller $\tau$ step with the same ratio. For the baselines, we treat the forward timestep of each hidden layer as the same and extract their object position information as the prediction. Table 5 reports the mean and standard deviation of each setting. From Table 5 we can observe that:

- Clearly, SEGNO outperforms all other baselines across all settings by a large margin. Notably, when there is a lack of supervised signals at 250/500/750ts, the performance of all other baselines decreases significantly. By contrast, SEGNO achieves similar results as in 1000ts, demonstrating its robust generalization to short-term prediction.

- Another interesting point is that SEGNO's error exhibits a distinct trend compared to other baselines. While the errors of other baselines significantly increase with decreasing time steps, SEGNO achieves even smaller errors with shorter time steps. This observation justifies our theoretical results that the error is bound by the chosen timestep.

- Additionally, the standard deviation of SEGNO is much smaller than that of other baselines, indicating the numerical stability of SEGNO. This result further confirms our theoretical finding that SEGNO can obtain a better latent trajectory between two discrete states.

## C.2 COMPARISON WITH GNS

We conducted additional evaluations of GNS and SEGNO-avg., which are learned by minimizing average acceleration, on two N-body systems. The results of these evaluations are presented in Table 6. We can observe that SEGNO outperforms both GNS and SEGNO-avg. In all cases, showing that training second-order neural odes on position loss outperforms training models on average acceleration.

Table 6: Comparsion ($\times 10^{-2}$) between SEGNO and GNS on simulated N-body systems. Results averaged across 3 runs.

| Method | Charged | | | Gravity | | |
|---|---|---|---|---|---|---|
| | 1000 ts | 1500 ts | 2000 ts | 1000 ts | 1500 ts | 2000 ts |
| GNS | $3.245_{\pm 0.068}$ | $11.689_{\pm 0.330}$ | $31.632_{\pm 0.206}$ | $4.204_{\pm 0.081}$ | $17.095_{\pm 0.136}$ | $50.275_{\pm 0.201}$ |
| SEGNO-avg. | $2.146_{\pm 0.079}$ | $10.145_{\pm 0.034}$ | $24.244_{\pm 0.212}$ | $1.431_{\pm 0.047}$ | $19.488_{\pm 0.978}$ | $54.370_{\pm 1.385}$ |
| SEGNO | $\mathbf{0.433}_{\pm 0.013}$ | $\mathbf{1.858}_{\pm 0.029}$ | $\mathbf{4.285}_{\pm 0.049}$ | $\mathbf{0.338}_{\pm 0.027}$ | $\mathbf{1.362}_{\pm 0.077}$ | $\mathbf{4.017}_{\pm 0.087}$ |

## C.3 ROLLOUT COMPARISON ON N-BODY SYSTEMS

We evaluate the generalizability of models for rollout simulation. Specifically, we train all models on 1000ts and use rollout to make the prediction for the longer time step (over 40 rollout steps, indicating over 40000ts.). Figure 8 depicts the mean squared error of all methods on two datasets. We can observe that all baselines experience numerical explosion due to error accumulation during the rollout process, leading to a quick drop in prediction performance. In contrast, SEGNO demonstrates an order-of-magnitude error improvement over other baselines. This numerical stability can be attributed to the Neural ODE framework for modeling position and velocity.

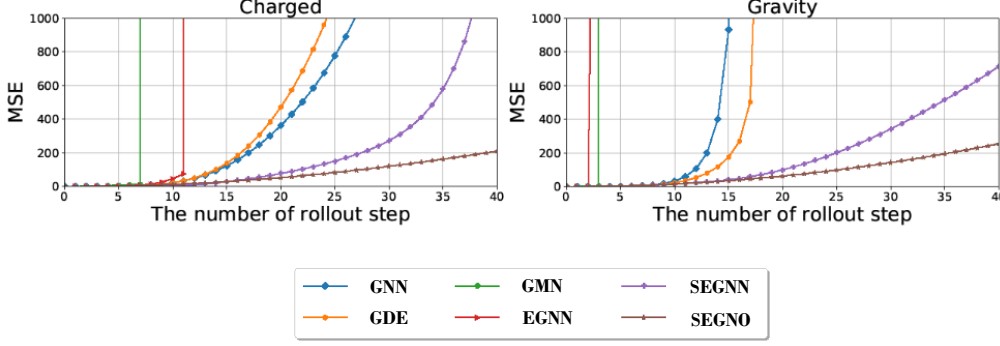

Figure 8: Mean squared error of rollout. Each rollout step is equal to 1000 ts. All models are trained on 1000ts.

## C.4 COMPARISON WITH PHYSICS-INFORMED GNNs

Following the experimental settings in (Thangamuthu et al., 2022), we thoroughly compare the rollout position, energy, and momentum errors of SEGNO with those of GNODE (Bishnoi et al., 2022) and HGNN (Thangamuthu et al., 2022) on N-body systems. GNODE and HGNN are trained by minimizing acceleration and Hamilton respectively. The target timestep is set to 100ts. The results are demonstrated in Table 7. We can observe that SEGNO consistently outperforms GNODE and HGNN in rollout errors. Although SEGNO is only trained on the system positions, it achieves comparable performance on energy and momentum errors with GNODE, which further validates its effectiveness.

Table 7: Rollout position, energy, and momentum errors. All models are trained on 100ts. Results averaged across 3 runs.

|  | Method | Charged | | | | | Gravity | | | | |
|---|---|---|---|---|---|---|---|---|---|---|---|
|  |  | 100 ts | 500 ts | 1000 ts | 1500 ts | 2000 ts | 100 ts | 500 ts | 1000 ts | 1500 ts | 2000 ts |
| Rollout | HGNN | 2.16 | 11.47 | 21.95 | 30.40 | 36.89 | 3.26 | 16.78 | 30.83 | 40.07 | 45.94 |
|  | GNODE | 0.36 | 5.41 | 14.05 | 21.61 | 27.39 | 0.32 | 6.42 | 17.53 | 26.17 | 32.16 |
|  | SEGNO | **0.06** | **4.26** | **12.26** | **19.42** | **25.08** | **0.04** | **5.62** | **16.84** | **25.71** | **31.82** |
| Energy | HGNN | 1.94 | 7.49 | 11.36 | 14.10 | 15.53 | 2.78 | 10.73 | 16.10 | 19.50 | 22.34 |
|  | GNODE | 0.82 | 5.82 | 10.87 | 14.07 | 15.78 | 0.68 | 7.46 | 14.40 | 18.39 | 20.85 |
|  | SEGNO | **0.79** | **5.74** | **9.93** | **12.66** | **14.05** | **0.11** | **7.38** | **13.58** | **16.94** | **18.89** |
| Momemtum | HGNN | 10.12 | 33.37 | 45.98 | 51.67 | 54.38 | 6.99 | 25.65 | 39.12 | 46.08 | 49.96 |
|  | GNODE | 4.17 | 28.65 | 43.97 | 50.84 | 54.06 | 2.35 | 21.43 | 36.93 | 44.92 | 49.19 |
|  | SEGNO | **2.17** | **28.39** | **43.81** | **50.46** | **53.54** | **0.75** | **21.40** | **36.88** | **44.65** | **48.42** |

## C.5 COMPARISON WITH ADVANCED FORCE FIELD PREDICTION MODELS

For a more comprehensive evaluation, we add empirical comparison on the MD22 dataset with NeuqIP (Batzner et al., 2022)[6] and Allegro (Musaelian et al., 2023)[7]. Both models take atom positions and numbers as input and optimize the errors between geometric outputs (generally treated as predicted forces) and true positions. We try our best to tune the models and the results are demonstrated in Table 8. It can be observed both methods do not perform well in predicting the time-dependent evolution of molecular dynamics, which is mainly attributed to the lack of dynamic information.

---

[6]https://github.com/mir-group/nequip
[7]https://github.com/mir-group/allegro/tree/main

Table 8: Comparison with equivariant GNNs in the domain of advanced force field prediction. Mean squared error ($\times 10^{-3}$) on MD22 dataset averaged across 3 runs are reported.

| Molecule | NequIP | Allgero | SEGNO |
|---|---|---|---|
| Ac-Ala3-NHMe | $12.060_{\pm 0.220}$ | $11.785_{\pm 0.193}$ | $\mathbf{0.418}_{\pm 0.038}$ |
| DHA | $13.275_{\pm 0.164}$ | $13.126_{\pm 0.121}$ | $\mathbf{0.370}_{\pm 0.020}$ |
| Stachyose | $11.375_{\pm 0.104}$ | $11.164_{\pm 0.135}$ | $\mathbf{0.440}_{\pm 0.096}$ |
| AT-AT | $9.178_{\pm 0.123}$ | $9.032_{\pm 0.125}$ | $\mathbf{0.548}_{\pm 0.006}$ |
| AT-AT-CG-CG | $8.959_{\pm 0.066}$ | $8.866_{\pm 0.125}$ | $\mathbf{0.394}_{\pm 0.033}$ |
| Buckyball Catcher | $5.418_{\pm 0.058}$ | $5.331_{\pm 0.077}$ | $\mathbf{0.199}_{\pm 0.020}$ |
| Double-walled Nanotube | $3.852_{\pm 0.077}$ | $3.794_{\pm 0.022}$ | $\mathbf{0.225}_{\pm 0.008}$ |

## C.6 RUNTIME COMPARISON ON N-BODY SYSTEMS

We evaluate the running time of each model on N-body systems with Telsa T4 GPU and report the average forward time in seconds for 100 samples. The results are listed in Table 9. We can observe that SEGNO's forward time (0.0227s) remains competitive compared to the best baseline SEGNN (0.0315s), indicating its efficiency.

Table 9: Forward time in seconds for a batch size of 100 samples on a Tesla T4 GPU.

| Linear | GNN | GDE | TFN | SE(3)-Tr. | RF | EGNN | GMN | SEGNN | SEGNO |
|---|---|---|---|---|---|---|---|---|---|
| 0.0002 | 0.0064 | 0.0088 | 0.0440 | 0.2661 | 0.0052 | 0.0126 | 0.0137 | 0.0315 | 0.0277 |

## C.7 GENERALIZATION CAPABILITY TO LARGE SYSTEMS

In this part, we evaluate the generalizability of models for larger system sizes. Specifically, we train all models on 5-body gravity systems and then test them on 10-body and 20-body gravity systems. Table 10 shows their results. We compare three strong baselines, i.e., EGNN, GMN, and SEGNN. The results show that the performance of all baselines significantly drops when testing on larger systems. In contrast, SEGNO still demonstrates a marked improvement over other baselines, especially on 20-body systems.

## C.8 MORE RESULTS ON CMU MOTION CAPTURE

This section illustrates more visualizations of GMN and SEGNO on modeling object motions. From Figure 9, we can observe that SEGNO is able to track the ground-truth trajectories accurately, which is consistent with the performance in Table 4.

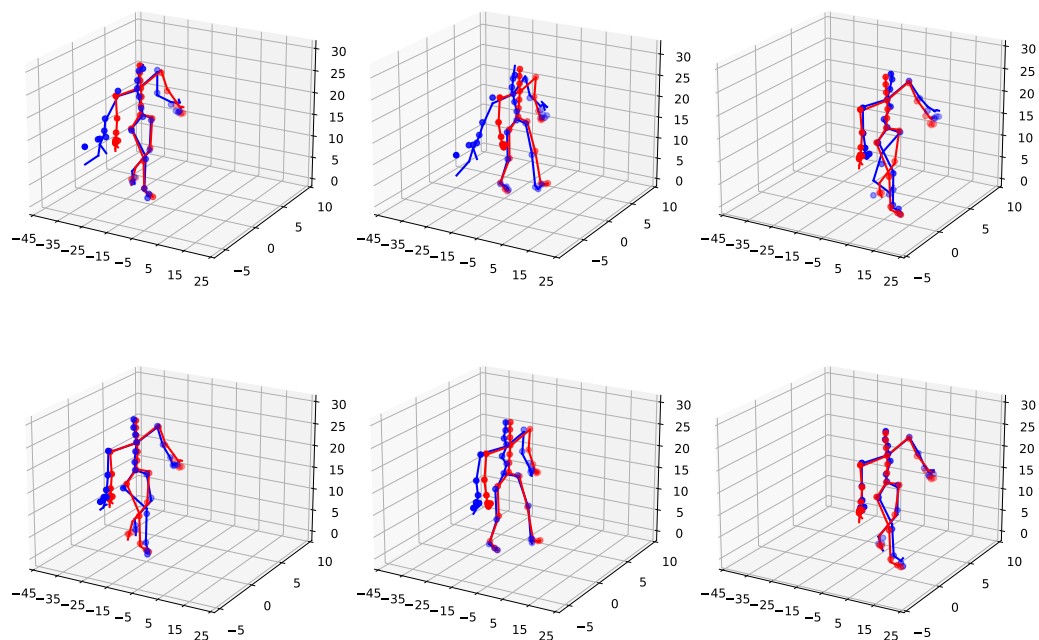

Figure 9: Example snapshots of Motion Capture with 50 time steps. *Top*: Prediction of GMN. *Bottom*: Prediction of SEGNO. Ground truth in red, and prediction in blue.

Table 10: The results on the larger systems. Mean squared error and the standard deviation are reported. Results are averaged across 3 runs.

| Method | 10-Body | 20-Body |
|--------|---------|---------|
| **EGNN** | $0.566_{\pm 0.316}$ | $1.985_{\pm 1.111}$ |
| **GMN** | $0.716_{\pm 0.314}$ | $\underline{1.323}_{\pm 0.430}$ |
| **SEGNN** | $\underline{0.333}_{\pm 0.036}$ | $3.937_{\pm 2.121}$ |
| **SEGNO** | $\mathbf{0.152}_{\pm 0.021}$ | $\mathbf{0.850}_{\pm 0.015}$ |

