# OpenReview forum: "SEGNO: Generalizing Equivariant Graph Neural Networks with Physical Inductive Biases"
_ICLR.cc/2024/Conference — ICLR 2024 spotlight_

### Official Review · Reviewer_mXJL · 2023-10-26

**Soundness:** 3 good
**Presentation:** 3 good
**Contribution:** 2 fair
**Rating:** 6
**Confidence:** 4

**Summary:**

## Summary

The paper investigates the integration of physical inductive biases, equivariant Graph Neural Networks (GNNs), and Neural Ordinary Differential Equations (ODEs) to improve model prediction accuracy. The authors validate their approach on 3 datasets: N-body, molecular dynamics, and captured human motion data. The technical contributions are sound though not groundbreaking. Given the paper's strengths and incremental contribution, I recommend a borderline acceptance.

## Detailed Comments

### Contributions

1. The paper shows that the error bound between the predictions and the ground truth outperforms previous methods focused on average accelerations.
2. It establishes that when the incremental function is either E(3) or O(3), the combined Neural ODE and equivariant GNN also possess these properties.
3. It shows enough empirical validation and ablation studies for later scholars to save their time and make proper choices.

### Comments and Suggestions

a) Lack of Visual Illustration: Apart from human motion data, the paper does not provide visual illustrations for any of the datasets. This makes it challenging to grasp the scope of the method's applicability, especially for readers not specialized in fields like molecular dynamics or N-body systems.

b) Figure 4 Inconsistency: The first column in Figure 4 is hard to interpret due to inconsistent color coding of initial conditions and predictions across different columns. It would be helpful to maintain color consistency.

c) Unclear experiment in Figure 2: It is unclear which experiment or dataset Figure 2 refers to, leading to possible confusion for the readers.

d) Inconsistent Sectioning for Human Motion Experiment: Unlike the other experiments, the human motion experiment is not provided a standalone section, making it confusing for readers.

e) Future Direction on Scalability: The paper could benefit from discussing scalability issues for large graphs. Since all examples in the paper involve graphs with fewer than 1,000 nodes, it would be interesting to explore the method's applicability to larger networks. Some relevant references include:

- [MultiScale MeshGraphNets](https://arxiv.org/abs/2210.00612)
- [Multi-GPU Approach for Training of Graph ML Models on large CFD Meshes](https://arxiv.org/abs/2307.13592)
- [Efficient Learning of Mesh-Based Physical Simulation with BSMS-GNN](https://arxiv.org/abs/2210.02573)
- [LazyGNN: Large-Scale Graph Neural Networks via Lazy Propagation](https://arxiv.org/abs/2302.01503)

## Conclusion

The paper presents a sound technical approach integrating physical inductive biases, equivariant GNNs, and higher-order Neural ODEs to improve model prediction accuracy. It's a sound but not groundbreaking paper. Therefore, I recommend a borderline acceptance for this paper.

**Strengths:**

See above

**Weaknesses:**

See above

**Questions:**

See above

---

> ### Author Response · Authors · 2023-11-17
> **Response to Reviewer mXJL**
>
> Thank you for the provided comments and helpful suggestions! We have enhanced our paper accordingly.
>
> >Comment a: Lack of Visual Illustration.
>
> Thanks for the instructive suggestion. We have included visualizations of N-body systems in the revised version.
>
>
> >Comment b: Inconsistent color in Figure 4.
>
> Thanks for the comment. We have corrected the colors of Figure 4.
>
> >Comment c: Figure 2 description.
>
> Thanks for raising this issue. Figure 2 is a case study of charged N-body systems where EGNN and SEGNO are trained to predict positions after 1000ts. We have added it to the Figure 2 caption in the revised version.
>
> >Comment d: Inconsistent Sectioning for Human Motion Experiment.
>
> We apologize for the confusion. We adjust section arrangements to enhance readability in the revised version.
>
> >Comment e: Future Direction on Scalability.
>
> Thank you for providing insightful future direction and valuable references! We agree that it is a very interesting point to explore our framework to solve the dynamics of large graphs. Analogous to our work, a feasible solution is to parameterize the target partial differential equation (PDE) via a GNN and incorporate solvers to train the model. We have added the references and will explore them in future work.

---

> ### Author Response · Authors · 2023-11-23
> **A Summary of Our Rebuttal**
>
> Thanks very much for your time and valuable suggestions. In the rebuttal period, we have added additional visualizations of N-body systems in Appendix, corrected Figure 4 colors, Figure 2 description, and inconsistent sectioning for the human motion experiment. We also appreciate the insightful future direction and valuable references. We have added it as future work in the revised version.
>
> Would you mind checking our responses and confirming whether you have any further questions?
>
> Any comments and discussions are welcome!
>
> Thanks for your attention and best regards.

---

### Official Review · Reviewer_rsLd · 2023-10-31

**Soundness:** 4 excellent
**Presentation:** 3 good
**Contribution:** 4 excellent
**Rating:** 8
**Confidence:** 3

**Summary:**

The authors present an innovative graph neural network framework that introduces second-order continuity into the modeling process. Unlike traditional discrete updates in other graph neural networks, they employ neural ordinary differential equations to capture continuous trajectories. This approach preserves the equivariance property of the underlying graph neural networks. Their model is rigorously evaluated across a range of standard test systems, including time evolution of N-body systems, molecular trajectories, and human motion trajectories. In all experiments, their model consistently outperform existing methods.

**Strengths:**

- The paper is well written and structured
- The authors introduce a novel framework for graph neural networks that incorporates second-order continuity by utilizing neural ODEs. This innovation enables more accurate modeling of physical systems, as many are governed by second-order laws.
- Their approach stands out by allowing the modeling of continuous trajectories, in contrast to existing graph neural networks that rely on discrete updates.
- They rigorously prove the claims they make, such as the preservation of equivariance and the derivation error bounds.
- Their model undergoes extensive testing on various time series prediction tasks, consistently outperforming previous models and showcasing its superior performance.
-  The insightful ablation study underscores the significance of incorporating second-order dynamics and utilizing a continuous model instead of a discrete one.

In conclusion, I consider this paper to be a significant and valuable addition to the community, and I recommend its acceptance.

**Weaknesses:**

- Run times for the various methods are not reported. Given the neural ODE nature of SEGNO, it's reasonable to assume that this model might have longer execution times compared to some of its competitors. It would be advantageous to include these run times in the appendix, with brief mentions in the main text for clarity.

**Questions:**

- Can SEGNO be included in a Markov Chain Monte Carlo (MCMC) framework aimed at accelerating Molecular Dynamics simulations, a common application for predicting future trajectory states? Typically, such applications require Metropolis-Hastings corrections to asymptotically sample from the equilibrium distribution.

- Graph neural networks are known for their transferability, as demonstrated by other methods on the QM9 dataset. While acknowledging the difference in tasks, do the authors envision the potential for SEGNO to be employed in a similarly transferable manner?

---

> ### Author Response · Authors · 2023-11-17
> **Response to Reviewer rsLd**
>
> We really appreciate the reviewer for listing the detailed strengths, which have clearly recognized the novelty and contributions of our paper. As requested by the reviewer, we have experimentally shown the running time of all models.
>
> >Weakness 1: Running time of models.
>
> Thanks for your suggestion! We evaluate the running time of each model on N-body systems with Telsa T4 GPU and report the average forward time in seconds for 100 samples. The results are listed below:
>
> | Model | Linear | GNN | GDE | TFN | SE(3)-Tr.| Radial Field | EGNN | GMN | SEGNN | SEGNO |
> | :---- | :---- | :---- | :---- | :---- | :---- | :---- | :---- | :---- | :---- | :---- |
> | Time(s)| 0.0002 | 0.0064 | 0.0088 | 0.0440 | 0.2661 | 0.0052 | 0.0126 | 0.0137 | 0.0315 | 0.0277 |
>
> We can observe that SEGNO's forward time (0.0227s) remains competitive compared to the best baseline SEGNN (0.0315s), indicating its efficiency.
>
> >Question 1: Can SEGNO be included in a Markov Chain Monte Carlo (MCMC) framework？
>
> The reviewer has raised an interesting question that would serve as a possible future direction. Although both SEGNO and the MCMC framework (Metropolis-Hastings) aim to explore molecular dynamics, they are different in the following aspects:
> * SEGNO primarily focuses on simulating the time-dependent evolution inherent in Newtonian dynamics.
> * MCMC framework is targeted towards thermodynamic analysis of molecular systems at equilibrium, rather than dynamic simulations of how systems evolve over time.
>
> Given their different operational domains and objectives in molecular dynamics, integrating SEGNO with the MCMC framework remains a challenging task. However, a possible solution is to replace the target ODE of SEGNO with a Stochastic Differential Equation (SDE). Thus, MCMC framework can be employed to estimate parameters within the SDE. We will explore it in the future.
>
> >Question 2: Do the authors envision the potential for SEGNO to be employed in a similarly transferable manner?
>
> Yes! We agree that it is a very interesting and promising point to explore. To investigate the model transferability across different systems, we train models on 5-body Gravity systems and test them on 10-body and 20-body systems. The goal is to predict the positions after 1000ts. The results are as follows:
>
> | System | EGNN | GMN | SEGNN | SEGNO |
> | :---- | :---- | :---- | :---- | :---- |
> | 10-body | 0.566 | 0.716 | 0.333 | $\textbf{0.152}$ |
> | 20-body | 1.985 | 1.323 | 3.937 | $\textbf{0.850}$ |
>
> We can observe that although SEGNO does not train on trajectories of 10-body and 20-body systems, it achieves significantly better performance, demonstrating its transferability across different systems.
>
> Moreover, for transferability across different tasks, since the goal of SEGNO is to model second-order dynamical systems (i.e., Eq.1), it can be naturally generalized to predict other dynamic states such as energy and momentum.

---

> > ### Comment · Reviewer_rsLd · 2023-11-21
> >
> > I appreciate the response and the inclusion of additional experiments. My initial rating remains unchanged, as I am content with the overall outcome.

---

> > > ### Author Response · Authors · 2023-11-22
> > >
> > > Thanks for your reply. We are encouraged by your positive comments.

---

### Official Review · Reviewer_GrTs · 2023-11-05

**Soundness:** 3 good
**Presentation:** 3 good
**Contribution:** 2 fair
**Rating:** 6
**Confidence:** 4

**Summary:**

In this work, authors combine neural ODE with equivariant GNN to propose a second order equivariant GNODE (SEGNO). Authors prove that the SEGNO maintains equivariance. The study prove that the discrepancy between the trajectory learned by SEGNO and the ground truth is bounded. Results show that the SEGNO provides relatively superior performance on various dynamical systems.

**Strengths:**

1. Learning dynamical systems with physics-based inductive bias is an important problem to be studied.
2. Combining equivariance with NODE is an interesting approach.
3. Several interesting and varied datasets are considered for empirical examples including n-body systems, MD22, and motion capture datasets.
4. The results show that the SEGNO outperforms baselines for several tasks including molecular dynamics, and motion capture.

**Weaknesses:**

1. The idea of combining GNODE with an equivariant GNN is fairly straightforward. In fact, there are several works on GNODE that employ a second-order bias (Gruver et al., Bishnoi et al., both cited in the paper). While the backbone GNN can have some effect on the GNODE (as shown in the work: Thangamuthu, A., Kumar, G., Bishnoi, S., Bhattoo, R., Krishnan, N.M. and Ranu, S., 2022. Unravelling the performance of physics-informed graph neural networks for dynamical systems. Advances in Neural Information Processing Systems, 35, pp.3691-3702.), there are no restrictions on employing an equivariant GNN backbone with GNODE. Thus, the novelty of the work may be limited.

2. In the present work, GMN is used as the backbone GNN along with the GNODE based inductive bias. The choice of the specific backbone may be appropriate for some datasets such as motion capture. However, the architecture does not perform well for molecular datasets or even n-body systems. Evaluation on additional backbone architectures is missing.

3. The baselines considered are simple and not necessarily SOTA. For the MD22, there are several equivariant GNNs such as DimeNet, NequIP, Allegro, MACE, BOTNet, Equiformer, etc. (and many more newer architectures) that have shown SOTA performance. It is not clear why such advanced baselines or graph architectures are not used.

4. The metrics used for comparison does not necessarily capture the complete picture regarding the dynamics. There are several metrics such as energy violation error, momentum error, rollout error etc. in the literature. Moreover, for atomic structures additional metrics such as JS divergence of the radial distribution functions capture how well the predicted the structure is similar to the ground truth. Such metrics have not been included in the present work leading to an incomplete evaluation.

5. There are several works on articulated rigid body in the literature including physics-informed GNNs such as Hamiltonian NNs, and Lagrangian graph neural networks for articulated body. Comparison with such baselines are not included in the work.

**Questions:**

1. Authors may rephrase the text ``accurately approximate'' in the proof of theorem 4.2?
2. Theorem 4.1 is a standard proof for any dynamical system and is not specifically related to SEGNO. It is not clear why this is a property of SEGNO and discussed under section 4. Authors may comment.
3. Proposition 4.2 is trivially dependent on the universal approximation theorem and is not needed to be stated as a proposition. Essentially, it proves that a NN can approximate the function $f(q,h)$ by minimizing the error between $q_{\theta}$ predicted by a NN and the ground truth $q$. This how every NN is trained. Authors may explain why this warrants a separate proof.
4. The present work focusses on combining equivariant GNNs with GNODEs. There are several ways of considering baselines. One set of baselines would be different equivariant architectures directly employing a data-driven approach to predict the dynamics. This is how some of the baselines are included in the present work. However, extensive literature shows that physics-informed inductive bias such GNODE, Hamiltonian GNN, and Lagrangian GNN can provide superior performance. Comparison with other such physics-informed GNNs such as Hamiltonian and Lagrangian GNN would have been insightful.
5. Another set of baselines would be GNODE with different equivariant backbone architectures. This has not been extensively evaluated in the present work. In the present work, authors employ GMN as the backbone. While it makes sense for the motion capture video, it is not clear why GMN would be a good backbone for the MD simulation dataset. Backbones such as NequIP would be better for such cases. Authors are encouraged to evaluate this.

---

> ### Author Response · Authors · 2023-11-17
> **Response to Reviewer GrTs - (1/3)**
>
> Thanks for your constructive comments! We try our best to address your concerns about the novelty and theoretical findings. As suggested by the reviewer,  extensive additional experiments including additional baselines, metrics, and backbones, are conducted to further support our contributions.
>
> ### On the novelty.
> >Weakness 1: The idea of combining GNODE with an equivariant GNN is fairly straightforward.
>
> Thanks for the comment. We would like to clarify that our contributions do **NOT** lie in the combinations of GNODE and equivariant GNNs. Existing studies[1,2] mainly employ a GNN that iteratively updates the latent representation to forecast the average acceleration. We distinguish our novelty and significance in the following aspects:
>
> * Our goal of providing theoretical insights to build Graph Neural ODE is novel and has never been explored before as far as we know. We show that a second-order Graph Neural ODE that iteratively updates positions instead of latent representations, can obtain bounded error of instantaneous acceleration and position through minimizing position discrepancy.
> * We thank the reviewer for raising the comparison with the works[1,2]. We validate our findings in novel applications (e.g., molecular and human motion dynamics) where the geometric property is crucial and propose incorporating equivariant GNNs to fulfill this inductive bias. Empirical results demonstrate our model has better generalization ability.
> * Thanks for your reference[1]. We agree that it is feasible to change NODE[1] backbones to equivariant GNNs. Nevertheless, it is still not clear whether such replacement affects the equivariant property of models. Our Proposition 3.1 provides a guarantee to build an equivariant GNODE.
>
>
> Thanks for your valuable references. We have added discussions of these papers in the revised version. We believe such a discussion can clarify the main contribution of our paper.
>
> [1] Unravelling the performance of physics-informed graph neural networks for dynamical systems. NeurIPS, 2022.
>
> [2] Enhancing the inductive biases of graph neural ode for modeling dynamical systems. ICLR, 2023.
>
> [3] Deconstructing the inductive biases of hamiltonian neural networks. ICLR, 2022.
>
> ### Compared with advanced force field prediction models, such as NequIP and Allegro.
>
> Thanks for your suggestions. We would like to clarify that the goal of baselines such as DimeNet, NequIP, and Allegro, is force field prediction, which is different from our task as follows:
> * **Force field prediction** is a static task that mainly takes atom positions as input, and outputs molecular energy and atom forces. To ensure the conservation of energy, the force predictions are partial gradients of predicted energy.
> * **Trajectory prediction** is a dynamic task that mainly takes atom positions and velocity as input, and outputs future atom positions.
>
> >Weakness 3: It is not clear why advanced baselines, such as DimeNet, NequIP, Allegro, MACE, BOTNet, Equiformer, etc., are not used.
>
> Thanks for your suggestion. We add empirical comparison on MD22 with official implementation [Nequip](https://github.com/mir-group/nequip)[1] and [Allegro](https://github.com/mir-group/allegro/tree/main)[2]. The models take atom positions and numbers as input and optimize the errors between geometric outputs (generally treated as predicted forces) and true positions. We try our best to tune the model and the results are as follows:
>
> | Molecule | NequIP | Allegro | SEGNO |
> | :----|:----|:----|:----|
> | Ac-Ala3-NHMe |  $12.060$ | $11.785$ | $0.779$|
> | DHA |  $13.275$ | $13.126$ | $0.887$ |
> | Stachyose |  $11.375$ | $11.164$ | $0.816$ |
> | AT-AT |  $9.178$ | $9.032$ | $0.501$ |
> | AT-AT-CG-CG |  $8.959$ | $8.866$ | $0.617$ |
> | Buckyball Catcher | $5.418$ | $5.331$ | $0.390$ |
> | Double-walled | $3.852$ | $3.794$ | $0.325$ |
>
>
> It can be observed they do not perform well in this setting due to the lack of dynamic information.
>
> >Question 5: Backbones such as NequIP would be better for such cases.
>
> We are grateful for advising us to consider NequIP as backbones, which is an important direction to improve model performance. However, as shown in the above results, it is non-trivial to incorporate dynamic geometric features (e.g., velocity) in the framework of irreducible representations. The ways that map velocity into high-order irreducible representations, design interaction modules, and decode dynamic states from irreducible representations are still an open problem. Thus, we are interested in exploring them in future works. We provide the results of the EGNN backbone in the following responses.

---

> ### Author Response · Authors · 2023-11-17
> **Response to Reviewer GrTs - (2/3)**
>
> ### Additional backbone.
>
> >Weakness 2: Evaluation on additional backbone architectures is missing.
>
> Thanks for the suggestion. We report the results of EGNN and GMN backbones as follows:
>
> | SEGNO Backbone | EGNN | GMN |
> | :----|:----|:----|
> | Ac-Ala3-NHMe |  $0.801$ | $0.779$|
> | DHA |  $0.909$ |  $0.887$ |
> | Stachyose |  $0.841$ |  $0.816$ |
> | AT-AT |  $0.525$ |  $0.501$ |
> | AT-AT-CG-CG |  $0.641$ |  $0.617$ |
> | Buckyball Catcher | $0.401$ |  $0.390$ |
> | Double-walled | $0.328$ | $0.325$ |
>
> We can observe that employing the GMN backbone achieves better results, which could be attributed to its multi-channel feature representation.
>
> ### Additional metrics and physics-informed GNN baselines.
>
> >Weakness 4: There are several metrics such as energy violation error, momentum error, rollout error etc. in the literature.\
> >Weakness 5 & Question 4: Comparison with physics-informed GNNs.
>
> Thank you for raising the comparison with physics-informed GNNs. As suggested by the reviewer, we thoroughly compare the rollout position, energy, and momentum errors of our SEGNO with those of GNODE and HGNN on N-body systems [1]. GNODE and HGNN aim to predict future acceleration and Hamiltonian respectively. All models are trained by minimizing the errors between the prediction and dynamic states (e.g., positions or acceleration) after 100 timesteps (a 0.001 second as 1 timestep). The results are shown as follows:
>
> On Charged N-body systems:
>
> | Rollout | ts=100 |  ts=500 |ts=1000 | ts=1500 | ts=2000|
> | :----|:----|:----|:----|:----|:----|
> | HGNN |  $2.16$|  $11.47$|$21.95$| $30.40$ | $36.89$|
> | GNODE |  $0.36$|  $5.41$|$14.05$| $21.61$ | $27.39$|
> | SEGNO | $\textbf{0.06}$|  $\textbf{4.26}$ |$\textbf{12.26}$| $\textbf{19.42}$ | $\textbf{25.08}$ |
>
> | Energy | ts=100 |  ts=500 |ts=1000 | ts=1500 | ts=2000|
> | :----|:----|:----|:----|:----|:----|
> | HGNN |  $1.94$|  $7.49$|$11.36$| $14.10$ | $15.53$|
> | GNODE |  $0.82$|  $5.82$|$10.87$| $14.07$ | $15.78$|
> | SEGNO | $\textbf{0.79}$|  $\textbf{5.74}$ |$\textbf{9.93}$| $\textbf{12.66}$ | $\textbf{14.05}$ |
>
> | Momentum | ts=100 |  ts=500 |ts=1000 | ts=1500 | ts=2000|
> | :----|:----|:----|:----|:----|:----|
> | HGNN |  $10.12$|  $33.37$|$45.98$| $51.67$ | $54.38$|
> | GNODE |  $4.17$|  $28.65$|$43.97$| $50.84$ | $54.06$|
> | SEGNO | $\textbf{2.17}$|  $\textbf{28.39}$ |$\textbf{43.81}$| $\textbf{50.46}$ | $\textbf{53.54}$ |
>
> On Gravitational N-body systems:
> | Rollout | ts=100 |  ts=500 |ts=1000 | ts=1500 | ts=2000|
> | :----|:----|:----|:----|:----|:----|
> | HGNN |  $3.26$|  $16.78$|$30.83$| $40.07$ | $45.94$|
> | GNODE |  $0.32$|  $6.42$|$17.53$| $26.17$ | $32.16$|
> | SEGNO | $\textbf{0.04}$| $\textbf{5.62}$ |$\textbf{16.84}$| $\textbf{25.71}$ | $\textbf{31.82}$ |
>
> | Energy | ts=100 |  ts=500 |ts=1000 | ts=1500 | ts=2000|
> | :----|:----|:----|:----|:----|:----|
> | HGNN |  $2.78$|  $10.73$|$16.10$| $19.50$ | $22.34$|
> | GNODE |  $0.68$|  $7.46$|$14.40$| $18.39$ | $20.85$|
> | SEGNO | $\textbf{0.11}$| $\textbf{7.38}$ |$\textbf{13.58}$| $\textbf{16.94}$ | $\textbf{18.89}$ |
>
> | Momentum | ts=100 |  ts=500 |ts=1000 | ts=1500 | ts=2000|
> | :----|:----|:----|:----|:----|:----|
> | HGNN |  $6.99$|  $25.65$|$39.12$| $46.08$ | $49.96$|
> | GNODE |  $2.35$|  $21.43$|$36.93$| $44.92$ | $49.19$|
> | SEGNO | $\textbf{0.75}$| $\textbf{21.40}$ |$\textbf{36.88}$| $\textbf{44.65}$ | $\textbf{48.42}$ |
>
> We can observe that SEGNO consistently outperforms GNODE and HGNN in rollout errors. Although SEGNO is only trained on the system positions,  it achieves comparable performance on energy and momentum errors with GNODE, which further validates our findings.
>
> [1] Unravelling the performance of physics-informed graph neural networks for dynamical systems. NeurIPS, 2022.

---

> ### Author Response · Authors · 2023-11-17
> **Response to Reviewer GrTs - (3/3)**
>
> ### On the theoretical findings.
> In the following, we provide additional details to eliminate the confusion. It's important to note that these details do **Not** affect the correctness or contributions of our theoretical findings.
>
>
> >Question 2: Why is Theorem 4.1 a property of SEGNO?
>
> Apologize for the confusion. We agree that Theorem 4.1 is a general statement regarding the uniqueness of solutions for dynamic systems. This statement serves as the prerequisite for Proposition 4.2 that SEGNO is capable of approximating the instantaneous acceleration of such a unique trajectory. We acknowledge that the current section title may potentially lead to misunderstandings. Thus we change the title as follows:
> * Section 4 -> SEGNO Analysis
> * Theorem 4.1 -> Lemma 4.1
>
>
> >Question 1: Authors may rephrase the text `accurately approximate` in the proof\
> Question 3: Why Proposition 4.2 warrants a separate proof?
>
>
> We apologize for the confusion caused by the omission of some technical details in the proof. The non-trivial aspects of the proof mainly lie in demonstrating that minimizing the position loss $|| \mathbf{q}\_{\theta}^{(t\_{1})} - \mathbf{q}^{(t\_{1})} ||\_p$ can enable neural networks to approximate the acceleration $f(\mathbf{q}^{(t)})$. Previous studies either directly predict the tartget position $\mathbf{q}^{(t\_{1})}$ by minimizing position loss $|| \mathbf{q}\_{\theta}^{(t\_{1})} - \mathbf{q}^{(t\_{1})} ||\_{p}$[1,2] or approximate acceleration $f(\mathbf{q}^{(t)}, \mathbf{h})$ using acceleration loss $|| f\_{\theta}(\mathbf{q}\_{\theta}^{(t)}) - f(\mathbf{q}^{(t)})||\_{p}$[3].
>
> In contrast, SEGNO has the ability to convert position loss into acceleration loss. In the meantime, it's a common practice to leverage the universal approximation theorem to establish the universal approximation capability of NNs within deep learning community[4,5,6]. Therefore, according to the aforementioned analysis and universal approximation theorem, it can be shown that SEGNO is capable of approximating acceleration $f(\mathbf{q}^{(t)}, \mathbf{h})$ with position loss.
>
> Due to the space constraints, we provide only a concise discussion here. For the complete mathematical derivation, we kindly refer the reviewer to the proof presented in the revised version. Additionally, we would like to re-emphasize that, while Proposition 4.2 seems intuitive, this property has made significant contributions in the following aspects:
> * To the best of our knowledge, it's the first theoretical justification that NNs equipped with SEGNO framework possess the capability to approximate the unique acceleration using position loss.
> * Empirically, it enables SEGNO to recover latent trajectories between input and output states, as illustrated in Fig. 2.
>
>
> Regarding your concern about the term ``accurately approximate``, we agree that it introduces ambiguity pertaining to the level of accuracy. To address the vagueness, we'll rephrase the text to replace ``accurately approximate`` with simple ``approximate``.
>
> [1] Vıctor Garcia Satorras, Emiel Hoogeboom, and Max Welling. E (n) equivariant graph neural networks. ICML, 2021.
>
> [2] Wenbing Huang, Jiaqi Han, Yu Rong, Tingyang Xu, Fuchun Sun, and Junzhou Huang. Equivariant graph mechanics networks with constraints. ICLR, 2022.
>
> [3] Alvaro Sanchez-Gonzalez, Jonathan Godwin, Tobias Pfaff, Rex Ying, Jure Leskovec, and Peter Battaglia. Learning to simulate complex physics with graph networks. ICML, 2020.
>
> [4] Universal Invariant and Equivariant Graph Neural Networks. NeurIPS, 2019.
>
> [5] A Universal Approximation Theorem of Deep Neural Networks for Expressing Probability Distributions. NeurIPS, 2020.
>
> [6] Deep network approximation: achieving arbitrary accuracy with fixed number of neurons. JMLR, 2022.

---

> ### Author Response · Authors · 2023-11-23
> **A Summary of Our Rebuttal**
>
> Thanks very much for your time and valuable comments. As the discussion will close in a few hours, we would be grateful if you could allocate some time to review our response.
>
> We understand that you have a multitude of responsibilities, thus **we have summarized our response** as follows:
>
>
> * We add discussions of existing GNODEs to clarify the main contribution of our paper ([Response-(1/3)](https://openreview.net/forum?id=3oTPsORaDH&noteId=U0SOHVBnvS)).
> * We add advanced force field prediction models as baselines ([Response-(1/3)](https://openreview.net/forum?id=3oTPsORaDH&noteId=vsqUg3gkFt)).
> * We empirically compare our model with physics-informed GNN baselines w.r.t. the rollout position, energy, and momentum errors ([Response-(2/3)](https://openreview.net/forum?id=3oTPsORaDH&noteId=vsqUg3gkFt)).
> * We evaluate the performance of different backbones ([Response-(2/3)](https://openreview.net/forum?id=3oTPsORaDH&noteId=vsqUg3gkFt)).
> * We provide additional details to eliminate the confusion about our theoretical findings ([Response-(3/3)](https://openreview.net/forum?id=3oTPsORaDH&noteId=t4JsxUtzn1)).
>
> Would you mind checking our responses and confirming whether you have any further questions?
>
> Any comments and discussions are welcome!
>
> Thanks for your attention and best regards.

---

> > ### Comment · Reviewer_GrTs · 2023-12-02
> > **Thank you**
> >
> > I thank the authors for the detailed clarifications and additional experiments. I have now increased the score.

---

### Author Response · Authors · 2023-11-21
**General Response to All Reviewers**

Dear Reviewers,

We sincerely thank all the reviewers (GrTs,rsLd,mXJL) for their valuable feedback. We are glad that the reviewers appreciated the significance of our problem (GrTs), the interest and novelty of our proposed framework (GrTs, rsLd), the soundness of our theoretical analysis (mXJL, rsLd), the comprehensiveness of our experiments (rsLd, mXJL), and the overall quality of our paper's writing (rsLd).


We have made every effort to faithfully address your comments in the responses. As suggested by the reviewers, we add

* Additional experiments of physics-informed GNN baselines w.r.t. the rollout position, energy, and momentum errors ([GrTs](https://openreview.net/forum?id=3oTPsORaDH&noteId=vsqUg3gkFt)).
* Additional results of different backbones ([GrTs](https://openreview.net/forum?id=3oTPsORaDH&noteId=vsqUg3gkFt)).
* Additional empirical comparisons with advanced force field prediction models, including NequIP and Allegro ([GrTs](https://openreview.net/forum?id=3oTPsORaDH&noteId=U0SOHVBnvS)).
* Additional evaluations on model efficiency and system generalization ([rsLd](https://openreview.net/forum?id=3oTPsORaDH&noteId=KXOyPj2eUc)).
* Additional theoretical details for Lemma 4.1 and Proposition 4.2 ([GrTs](https://openreview.net/forum?id=3oTPsORaDH&noteId=t4JsxUtzn1)).
* Visualizations of N-body systems in Appendix,  corrected Figure 4 colors, Figure 2 description, and inconsistent sectioning for human motion experiment ([mXJL](https://openreview.net/forum?id=3oTPsORaDH&noteId=CY1mQEsR3z)).
* Referenced studies from all reviewers in the related works section.


We have incorporated the suggested modifications in the revised version, which are highlighted in blue. As the deadline for discussion (closes on Nov 22) is fast approaching, we would be grateful if you could allocate some time to review our responses.

Thanks for all the reviewers' time again.

Best regards,

Authors

---

### Comment · Area_Chair_wdM1 · 2023-11-21
**Reviewers: Please respond to authors or update review**

Dear Reviewers,

The discussion phase will end tomorrow.  Could you kindly respond to the authors rebuttal letting them know if they have addressed your concerns  and update your review as appropriate? Thank you.

-AC

---

> ### Author Response · Authors · 2023-11-22
>
> Dear Area Chair,
>
> Thanks for your kind reminder.
>
> Best regards,
>
> Authors

---

### Meta-Review · Area_Chair_wdM1 · 2023-12-09

**Metareview:**

This paper proposes Second-order Equivariant Graph Neural Ordinary Differential Equation (SEGNO) a method for modeling dynamics by computing second derivates with equivariant GNNs. The authors analyze the methods expressivity and error theoretically and empirically demonstrate the method on n-body problems, molecular dynamics, and motion capture.

Reviewers found this to be an interesting approach, combining second-order neural ode with equivariant GNNs, to an important problem, modeling complex dynamics. The paper contains varied experimental settings which give convincing evidence of the models value. The model outperforms a variety of baselines including other equivariant GNNs and Hamiltonian NNs with respect to domain relevant metrics. Moreover, the authors provide extensive theoretical analysis of their model including proofs of equivariance, uniqueness of output trajectory, and error bounds.  The method can model continuous trajectories and by modeling instantaneous acceleration achieves lower error than previous second order graph neural ode methods.

**Justification For Why Not Higher Score:**

- the method is novel but is potentially not a huge leap given that two second order graph neural ode method existed and many flavors of equivariant GNN existed.
- integration with other SOTA equivariant GNN backbones remains future work which could potentially further boost model performance

**Justification For Why Not Lower Score:**

+ novel method for important problem
+ extensive experiments across several domains with strong baselines and domain relevant metrics
+ rigorous theoretical analysis of method

---

### Decision · Program_Chairs · 2024-01-16

Accept (spotlight)